behaviour, evolution

mating systems, monogamy, social behaviour, familiarity, fitness, animal tracking

**Author for correspondence:**
Antica Culina
e-mail: a.culina@nioo.knaw.nl

# Familiarity breeds success: pairs that meet earlier experience increased breeding performance in a wild bird population

Antica Culina[1,2], Josh A. Firth[2,3] and Camilla A. Hinde[2,4]

[1]Department of Animal Ecology, Netherlands Institute of Ecology, Wageningen, The Netherlands
[2]Edward Grey Institute, Department of Zoology, University of Oxford, Oxford, UK
[3]Merton College, University of Oxford, Oxford, UK
[4]Behavioural Ecology Research Group, Department of Biology, Anglia Ruskin University, Cambridge, UK

AC, 0000-0003-2910-8085; JAF, 0000-0001-7183-4115; CAH, 0000-0001-9376-4023

In socially monogamous animals, including humans, pairs can meet and spend time together before they begin reproduction. However, the pre-breeding period has been challenging to study in natural populations, and thus remains largely unexplored. As such, our understanding of the benefits of mate familiarity is almost entirely limited to assessments of repeated breeding with a particular partner. Here, we used fine-scale tracking technology to gather 6 years of data on pre-breeding social associations of individually marked great tits in a wild population. We show that pairs which met earlier in the winter laid their eggs earlier in all years. Clutch size, number of hatched and fledged young, and hatching and fledging success were not influenced by parents' meeting time directly, but indirectly: earlier laying pairs had larger clutches (that also produce higher number of young), and higher hatching and fledging success. We did not detect a direct influence of the length of the initial pairing period on future mating decisions (stay with a partner or divorce). These findings suggest a selective advantage for a new pair to start associating earlier (or for individuals to mate with those they have known for longer). We call for more studies to explore the generality of fitness effects of pair familiarity prior to first breeding, and to elucidate the mechanisms underlying these effects.

## 1. Introduction

Repeated breeding with the same partner (i.e. pair fidelity) has been shown to benefit fitness through increased breeding success [1], and greater survival of pair members [1–3]. The increase in breeding success partly arises because of the 'mate familiarity effect' [1,4,5], even when accounting for confounding factors, such as age/experience related increases in reproductive parameters [1,6], as familiar partners improve coordination, cooperation and responsiveness [7,8]. Furthermore, shared vigilance and increased competitiveness (e.g. access to food or roosting sites) of a pair outside of the breeding context [5,9,10] probably benefit survival. These fitness benefits of mate familiarity play a role in the evolution of long-term partnerships and social monogamy in birds [1].

However, the concept of mate familiarity extends beyond that of repeated breeding within a pair [11]. Pre-breeding familiarity, even when based on short encounters of potential future partners (e.g. vole species which use short-term olfactory cues [11]) and in species that do not form a pre-breeding bond, has been shown to influence mate choice (e.g. studies on rodents, see overview in [11], birds, e.g. [12], or fish [13]). In species where new pair bonds form a relatively long, and variable amount of time before the subsequent mating event, as recently demonstrated in birds [14,15] familiarity can have further implications for fitness. Partners that meet earlier might have more time to assess

**Figure 1.** A schematic of the possible mechanisms through which time when a pair met prior to their first breeding attempt can influence breeding success (i.e. components of breeding success other than laydate) and later mating decisions (i.e. breed with the same partner, or divorce a partner). Full arrows represent direct, and dashed arrows indirect (i.e. acting through one or more other mediators) effects.

each other and thus make a more informed choice on whether to breed together [16,17], and will also have more time to develop compatible behaviours that increase breeding success (see [18]). In this way, familiarity can benefit individuals' breeding success [19] and survival through direct benefits (early pairing benefits, [16]). On the other hand, early pairing can also potentially reduce an individuals' fitness, for example, owing to decreased opportunities to sample more potential partners [15], and increased effort reinforcing the pair bond, which could be costly.

Although the fitness implications of meeting a breeding partner earlier are important from an evolutionary, behavioural and conservation perspective, only a few studies have acknowledged this possibility [8,11,12,13,16,17] and only two studies have addressed the link between pre-breeding familiarity and breeding success, both CONDUCTED in captive experimental setting [19,20]. The first one [20], conducted on a species with an unknown social mating system and probably only maternal care, showed that captive pygmy rabbits (*Brachylagus idahoensis*) which were familiar with their breeding partner through being housed in nearby cages raised more litters with more offspring. The second study [19], conducted on a socially monogamous biparental species, showed that the length of time that randomly matched pairs of captive bearded reedlings (*Panurus biarmicus*) spend together prior to breeding influenced their breeding success. Pairs with the longest pair formation period achieved the highest behavioural synchronization in nest building, bred the earliest, and had the highest hatching and fledging success [19]. Similar effects should also appear in free-living populations, especially if earlier pair formation allows for earlier breeding, which in turn increases breeding success (laydate has been shown to be under strong selection in many populations as it strongly affects subsequent components of breeding success, e.g. [21–24]).

We studied whether meeting a future breeding partner earlier translates into fitness benefits in a free living population of socially monogamous bird; the great tit (*Parus major*). We used radio-frequency identification (RFID) technology to track wild individuals over the pre-breeding winter period (when breeding pairs form [15,25]), and we monitored their reproduction in the following spring. Specifically, we tested whether breeding success of a pair can be predicted from the time when a pair was first detected together on feeders in winter. Furthermore, we tested whether meeting time affects components of breeding success directly (likely via an increased coordination of parental duties of familiar pairs), and/or indirectly (e.g. birds that meet earlier have an earlier laydate, which further relates to reproductive benefits) as represented in figure 1. Previous studies have already shown that earlier laydate generally translates into higher breeding success and reproductive output [21–24].

Finally, we tested a prediction that pairs which meet earlier in winter have a higher probability of breeding together again in the following year (i.e. breeding together twice). We expect to find this if birds that meet earlier have a longer period of partner assessment (direct route, figure 1) or if pairs that meet earlier have increased breeding success, which in turn reduces divorce probability (indirect route, figure 1). Our study system is particularly well suited for answering these questions as birds of future mating pairs associate more consistently and generally more frequently with each other in winter flocks compared to other flock-mates [25]. Furthermore, previous work [15] has shown that meeting time is a good representation of the onset of pair bonding in this study system. We discuss the wider implications of our findings (meeting a future partner earlier on has fitness benefits in a wild population) and the broader significance of pair familiarity that extends beyond the breeding season.

## 2. Methods

### (a) Study system

The great tit is a small, short-lived (adult annual survival rates 25–65%, mean lifespan 1.9 years), cavity-nesting passerine [26]. Pairs are socially monogamous and breed on distinct territories, laying on average 8.5 eggs, and fledge on average seven chicks [24,26] per nest. After the breeding season, birds form fission–fusion flocks

[27–29] and breeding pairs generally form between birds co-occurring together in the same winter flocks [15,25,30]. Most of the birds that will ever breed recruit into the breeding population in their first year, and around 14% of fledged young are recorded to go on to breed (i.e. are 'recruited') [31]. Reported between-year divorce rates in great tits are quite variable between populations, ranging between 0 and 51% (e.g. [31,32]).

Our data come from a wild population of great tits in Wytham Woods, Oxford (51° 46′ N, 1° 19′ W) that breeds almost exclusively in nest boxes ($n = 1020$ nest boxes, around 450 pairs $yr^{-1}$), between April and July, and close to 98% of pairs have only one breeding attempt per season. The majority of the breeders (53% of females and 60% of males) were born in Wytham [33]. The population is non-migratory, and the majority of breeders stay in the woods during winter [34]. Breeding data are collected using standard protocols [29,35] to record parental identities and the breeding parameters of a pair (date of first egg, maximal clutch size, number of hatched chicks and number of fledged young). The data on winter flocks were collected between the winter beginning in 2007 until the end of the winter in 2014. In our population, an average flock size is 7.5 individuals (range 2–80 birds) while the flock size experienced by an average individual is around five birds [28].

## (b) Winter data collection set-up

Data on winter flocks were collected using RFID technology: birds marked with passive integrated transponders (PIT-tags) were detected on feeders equipped with RFID-antennae [27–29], with 4023 unique great tits detected over the six study years considered here. In this study system, an estimated 82% of the primary population is marked with PIT-tags and visit the recording devices over the winter season [34]. The winter data collection protocol (spatial positions and opening times of feeders) changed halfway through the study. In the first three winters (2007/2008 to 2009/2010), data were collected between August and March when 16 (out of 67 feeders) were always opened, and these were rotated every 4 days. In the second three winters (2011/2012 to 2013/2014), data were collected between early September and early March when all of the 65 feeders were opened during each weekend. Detailed description of the data collection set-up can be found in [27–29,36]. The feeder-availability protocol minimized the possibility that flocks would get attracted to the constant food sources and we assume that data gathered at feeders represent snapshots of the social composition of different flocks at the time of recording. Furthermore, individuals' use of feeders is not related to reproductive success [29].

## (c) Meeting time and social structure

We applied the Gaussian mixture model for event streams method [25] to the spatio-temporal datastream collected from the loggers to determine flocking events [25,27]. This method is robust for inferring flocking events and preferable to other methods such as using arbitary time intervals [27]. Furthermore, the flocking events extracted using this method are known to be non-random in social composition [28], related to individuals social associations in other contexts [37,38] and important to various social processes such as information spread [39,40] and mating [15,25,36]. We defined pair meeting time as the month (the first three winters) or the weekend (the last three winters) when the members of the future breeding pair were first identified in the same flock (i.e. gambit of the group approach, [41]). Time-frames for these were chosen based on the collection set-up in each set of winters.

A typical individual in this dataset experiences approximately five flock mates [28] Thus, it is reasonable to assume that future breeding partners have indeed met in the foraging flock when first observed together. Furthermore, we know that birds of new

pairs tend to spend the prior winter in the same flocks together [25], and previous work on the same dataset has shown that calculating the time when a pair first met in this way is good approximation of the beginning of pair bonding and relates to future pairing behaviour [15]. Thus, we can be confident that the majority of pairs not only meet but also begin bonding in the time period (month, weekend) in which they were first detected in the same flock. Nevertheless, we also carried out additional sensitivity analysis to ensure that our investigations are robust to these assumptions (see the electronic supplementary material, Methods and statistical analysis section).

Social networks were constructed based on flocking events using a simple ratio index (SRI, [42]), that describes the affinity of two individuals to co-occur in the same flock, and can range from 0 (never observed together) to 1 (always observed together). SRI is calculated as the number of times (within a certain period: winter, month, weekend in our case) that two individuals were observed in the same flocking event together, divided by the sum of times when they were observed at all (in the same flock, in different flocks at the same time, and when only one of the individuals was observed). Our previous work has demonstrated that our data collection methods (sampling snapshots of social structure throughout the winter) in combination with creating SRI social networks from the underlying gambit-of-the-group approach are good representations of social associations and bonds between individuals [15,27].

## (d) Dataset construction

To be included in our primary analysis, a pair had to meet several requirements. First, it had to be newly formed in the winter prior to the breeding season. This excludes pairs that had previously bred together. Second, both pair members had to be detected at winter feeders at least once. Third, both members had to be tagged either prior to the winter of interest (as nestlings or adults), or newly tagged in the current winter as immigrants at least two sampling periods before the pair meeting time. This eliminates pairs that had been associating but not detected as such purely because one or both members were not tagged at the time. Finally, we excluded six pairs (out of overall close to 400 pairs, i.e. less than 2%), where both individuals were known to be tagged prior to the winter, but were detected for the first time in the late autumn/winter (i.e. after October) as already paired. These pairs probably formed outside of the main woods, and returned already paired. These requirements were met by 169 pairs in 2008–2010 breeding seasons, and by 252 pairs of the 2012–2014 breeding seasons. Among these, members of 29, and nine pairs, respectively, were never detected in the same flock. In all of these pairs at least one of the birds was either detected at feeders very rarely (in the lowest 20th percentile of the distribution of the number of times each bird was detected in a given winter), or had a larger number of detections but in one month/weekend only. Thus, the final sample size for the analysis included 140 pairs breeding in 2008–2010, and 243 pairs breeding in 2012–2014 spring.

## (e) Statistical analysis

We ran separate sets of analysis for the two periods with a different protocol for winter data collection (2007/2008 to 2009/2010 winters; and 2011/2012 to 2013/2014 winters). We conducted statistical analysis in R [43] v.3.6.3 using package lme4 [44], and produced figures using package ggplot2 [45]. We compared the performance of the candidate models (please see below) based on the Akaike information criteria (AIC) values [46]. We considered that a model gained better support if its AIC value was 2 or more units lower than the AIC value of the competing model/models. For models that gained similar support (within 2 AIC), we considered the model with a fewer parameters to be more informative [47].

## (i) Meeting time and breeding success

To test whether meeting time can predict subsequent reproductive success, we ran separate model selection for each breeding success component. Our main response variables were: standardized laydate (date when the first egg was layed in the nest, standardized per year), clutch size (maximal number of eggs in the nest), the number of hatched young, the number of fledged young, hatching success (proportion of eggs that hatched), fledging success (proportion of hatchlings that fledged), and binary fledging success (fledged at least one young or not). We used generalized linear models (GLMs) with appropriate error distribution: Gaussian for laydate; Poisson for clutch size, number of hatchlings and number of fledglings; and binomial for hatching success and fledging success. In models with Poisson and binomial errors, and after model selection, we re-run the best model(s) with quasi-poisson and quasi-binomial error distribution to account for over-dispersion.

For each breeding success component, we ran the analysis in three steps (main analysis, table 1) and conducted a sensitivity analysis to confirm the robustness of the results (electronic supplementary material). When constructing our candidate models, we followed the rule that the number of estimated parameters ($k$) should be $k < $ 'sample size'/10 [48]. In the *first step* of model selection, we generated candidate models from combinations of three main predictors and their two-way interactions (table 1): meeting time, year (categorical variable) and pair type. Meeting time was a continuous variable, increasing monthly between 1 (August) and 8 (March) in winters 2007/2008 to 2009/2010 (eight values), and increasing weekly between 1 (weekend 2 of September) to 26 (weekend 1 of March) in winters 2011/2012 to 2013/2014 (26 values). Categorical variable *pair type* codes for males' and females' breeding experience in a population because previous work has showed that breeding experience can influence breeding success [49]. The variable can take four values: (i) both partners are experienced (previously bred in a population); (ii) both partners are new to the population (first year breeders and new immigrants); (iii) female is experienced, male is new; (iv) male is experienced, female is new. This way we have also partly controlled for age and immigration effects as 'new' breeders in our dataset are mostly immigrants and first-year breeders from the population. Breeding success components were correlated (see the electronic supplementary material, table S2 for the correlation coefficients). Thus, we controlled (additive effect) for: laydate when modelling clutch size, hatching and fledgling success; clutch size for the number of hatchlings and fledging success; and number of hatchlings when modelling the number of fledglings. This way we were also able to separate direct and indirect (i.e. acting through the preceding component of breeding success) effects of meeting time on breeding success (figure 1).

If a model (or models) with meeting time gained the best support (lowest AIC) in the first step, in the *second step*, we ran the same model with the quadratic and cubic effects of meeting time (table 1). We repeated the model selection on the number of fledglings and fledging success on a subset of data with pairs that had fledged at least one chick. We did this because complete fledging failure was often externally caused (e.g. predation), and thus does not fully reflect parental ability to raise chicks (22 pairs in the first study period, and 33 pairs in the second experience a complete fledging failure). We did not apply the same procedure to the hatching component because only one pair across years experienced complete hatching failure.

In our dataset, meeting time is, as expected, correlated with the male and female arrival time (please see the detailed analysis in the electronic supplementary material). Male and female arrival time were defined as the month (first three winters) or a weekend (last three winters) when a male or female was first detected in the dataset (same as the meeting time). Thus, if the

**Table 1.** The main steps of model selection to explore the influence of meeting time on breeding success. (In the first step, 16 generalized linear models with the listed basic structure were compared. The main predictor variables were meeting time (continuous variable), pair type (categorical variable with four categories) and year. If meeting time was supported as a predictor of breeding success, the quadratic and cubic relationship were modelled in the second step, and female and male arrival time in the third. $+$ = additive effect of a variable; $\times$ = interactive effect.)

| step 1 | model structure (for the explanatory variables) |
|---|---|
| | intercept |
| | pair type |
| | year |
| | meeting time |
| | year + meeting time × pair type |
| | year + meeting time + pair type |
| | year × meeting time + pair type |
| | meeting time + year × pair type |
| | meeting time × year |
| | meeting time + year |
| | meeting time × pair type |
| | meeting time + pair type |
| | year × pair type |
| | year + pair type |
| | year × pair type + meeting time × pair type |
| | year × pair type + meeting time × year |
| **step 2** | **modelling nonlinear effects** |
| | keep the structure of the best model(s) from step 1 and add quadratic and cubic function to the effect of meeting time |
| **step 3** | **test for the influence of male and female arrival time** |
| | (a) replace the meeting time with the female/male arrival time in the best model from above |
| | (b) control for female/male arrival time by adding it as an additive effect to the structure of the best model from above |
| **sensitivity analysis** | **check for the robustness of the results, based on the best model selected in the main analysis** |
| | (a) control for measures of the pair bond strength (additive effect) |
| | (b) control for preference for a partner (additive effect) |
| | (c) control for gregariousness (additive effect) |

previous steps supported meeting time as a predictor of breeding success, in the *third step* of the analysis, we checked whether: (a) male and female arrival time better explain variation in breeding success by replacing the term 'meeting time' in the best supported model from the previous steps, with male or female arrival time; and (b) adding male and female arrival time to the best model changes the estimate for the effect of the meeting time. Finally, we check the robustness of the results when controlling for several variables that describe the strength of the

pair association, preference for a partner, and gregariousness of individuals (details on this analysis are provided in the electronic supplementary material). We did not include a random term for the individual identity in models as less than 9% of individuals were replicated in this dataset.

### (ii) Future mating decisions: fidelity and divorce

We tested whether new pairs (first breeding together in spring $t$) that met earlier in winter (preceding $t$) had a higher probability of breeding together in breeding season $t + 1$ compared to pairs that started to associate later. We used data on only those pairs where both members had survived to breed in $t + 1$. We applied a GLM (with binomial error structure) with status of a pair in $t + 1$ as a binomial response (divorced or faithful), and meeting time as the main predictor. We controlled for breeding success, known to predict divorce probability in monogamous birds [50]. Specifically, we controlled for laydate and clutch size based on the results of a meta-analysis on breeding success and divorce that we have conducted using published data on great tits and three closely related species (please see details on meta-analysis in the electronic supplementary material, data used are a subset of data from [50]). To keep the models simple (owing to the small sample size; table 2), we only considered six possible structures for the explanatory part of the model for the first set of years (2007–2010, sample size is only 17 pairs): intercept model, 'clutch size', 'clutch size + meeting time', 'laydate', 'laydate + meeting time', 'meeting time'. For the second set of years (2011–2014), where the sample size was higher (50 pairs), we considered 12 possible models, six with the structure as outline above, and six where 'year' was additionally added as an additive effect.

## 3. Results

Pairs met throughout the winter in all 6 years ($n = 140$ pairs in 2007/2008 to 2009/2010; $n = 243$ pairs in 2011/2012 to 2013/2014, electronic supplementary material, figures S1 and S2). Some winters had a clear pair meeting peak at the very beginning (electronic supplementary material, figures S1 and S2). For the majority of pairs, the strength of the pair association in the meeting month/weekend (the bond meeting SRI value) was higher than the 75th percentile of all the social associations (i.e. other SRI values) of all the birds for that month/weekend (see the electronic supplementary material). Furthermore, for around 70% of future pairs, the pair association (measured as SRI) in the meeting month/weekend was among the 25% strongest social associations that a female and a male of a pair established in that month/weekend. Overall winter social association of the pair was higher for pairs that have met earlier in winter, then for those that have met later (see the electronic supplementary material, Predictors of pair association strength).

### (a) Meeting time and breeding success

Model selection on both datasets showed that meeting time had a direct effect on the standardized laydate (cubic term), and not on the other measures of breeding success (hatching and fledging success, number of eggs, hatchings or fledglings). Our results further showed that earlier laydate resulted in larger clutches, and in a higher hatching and fledging success. The parameter estimates of the best supported models are provided in the electronic supplementary material, tables S18–S21. The detailed model selection tables can be found in the electronic supplementary material, tables S3–S10 (for 2007/2008 to 2009/2010) and S11–S17 (for 2011/2012 to 2013/

**Table 2.** The number of pairs breeding together for the first time in the breeding season $t$, for which partners were tagged and detected associated in the winter prior to $t$, and where both survived to the season $t + 1$ to either breed together (fidelity) or with new partners (divorce). (For example, of six pairs that have formed in 2007/2008 winter (and bred for the first time in the 2008 breeding season), three divorced and three remained faithful to the 2009 breeding season.)

| status/ winter | 2007/ 2007 | 2008/ 2008 | 2009/ 2009 | 2011/ 2011 | 2012/ 2012 | 2013/ 2013 |
|---|---|---|---|---|---|---|
| divorced | 3 | 3 | 0 | 6 | 2 | 4 |
| faithful | 3 | 6 | 2 | 16 | 9 | 13 |

**Table 3.** Model summary (parameters estimates, standard errors, test statistics) of the best supported (i.e. lowest AIC) standardized laydate model for 2007/2008 to 2009/2010 dataset, and 2011/2012 to 2013/2014 dataset.

| parameter | estimate | s.e. | $t$-value | Pr(>|t|) |
|---|---|---|---|---|
| *2007/2008 to 2009/2010 dataset* | | | | |
| meeting time | 1.110 | 0.371 | 2.999 | 0.0032 |
| meeting time$^2$ | −0.217 | 0.090 | −2.405 | 0.0175 |
| meeting time$^3$ | 0.013 | 0.007 | 1.980 | 0.0497 |
| winter_07/08 | −1.489 | 0.452 | −3.291 | 0.0013 |
| winter_08/09 | −1.700 | 0.438 | −3.881 | 0.0001 |
| winter_09/10 | −2.002 | 0.470 | −4.263 | 3.78e-05 |
| *2011/2012 to 2013/2014 dataset* | | | | |
| intercept | −0.448 | 0.171 | −2.619 | 0.0093 |
| meeting time | 0.119 | 0.071 | 1.679 | 0.0945 |
| meeting time$^2$ | −0.012 | 0.007 | −1.801 | 0.0729 |
| meeting time$^3$ | 0.0004 | 0.0002 | 2.219 | 0.0274 |

2014 dataset). All of the results, and estimates of the effects, remained robust to our sensitivity analysis (please see the electronic supplementary material, Additional sensitivity analysis on the robustness of the results).

Estimates of the best-supported laydate model for 2007/2008 to 2009/2010 dataset (i.e. model with the lowest AIC, laydate$^3$ + year, table 3 for parameter estimates) showed that females which met their male in August laid their eggs the earliest (e.g. standardized laydate of −0.58, s.e. = 0.20 in 2008 spring). The laydate got later for females meeting their partner up to November (0.32, s.e. = 0.11), and stayed around the same value for females meeting their partners after November (figure 2$a$). Model selection provided no support for the effects of female or male arrival time on laydate (electronic supplementary material, table S3, third step).

The estimates of the laydate model with the lowest AIC (cubic effect of meeting weekend) for 2011/2012 to 2013/2014 dataset showed that females of pairs that met at the beginning of September laid the earliest (−0.34, 95% confidence interval (CI): −0.57/−0.11) and females of pairs that met at the beginning of March laid the latest (1.45, 0.84/2.05). However, the effect of meeting weekend was strongest after late December (i.e. the slope of the effect was steeper), figure 2$b$. There was no support that female or male arrival time predicted the laydate (electronic supplementary material, table S11, third step).

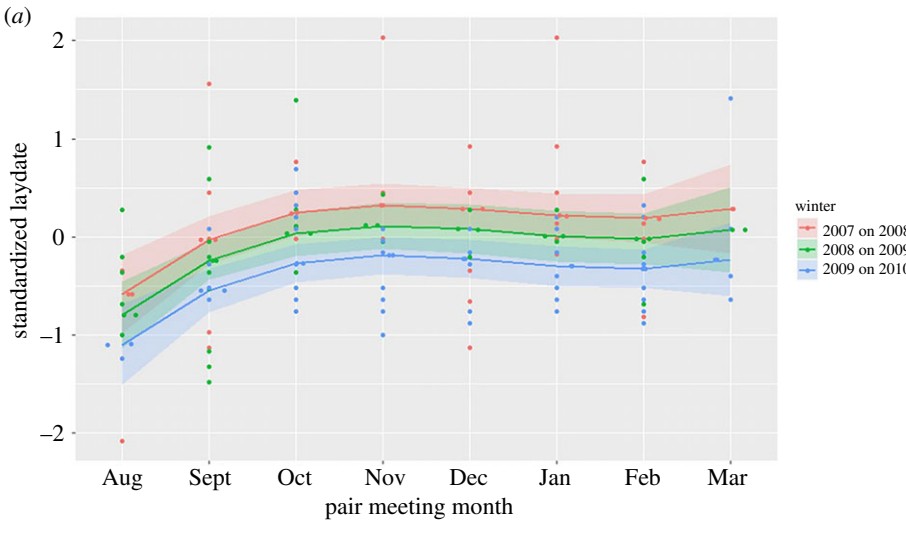

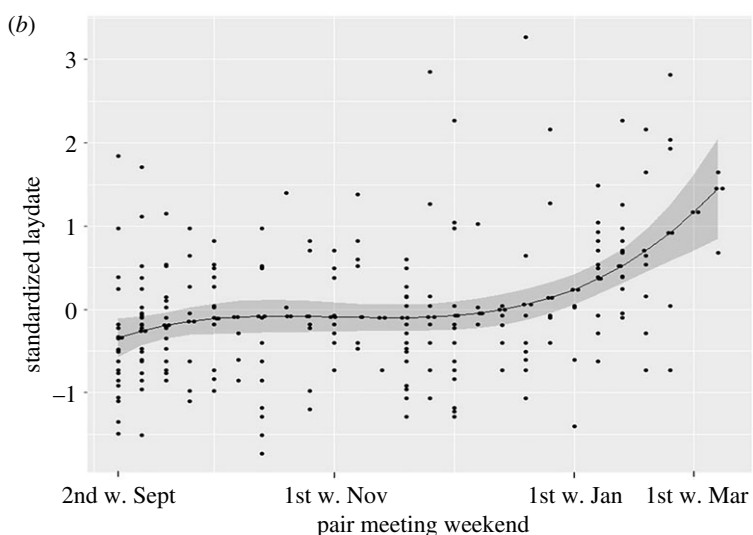

**Figure 2.** Standardized (for a year and for all breeding pairs) laydate of newly formed pairs of great tits as a function of meeting time of a pair in the preceding winter for (*a*) 2008 to 2010 breeding seasons, where meeting time was a month when a pair was first seen together; (*b*) 2012–2014 breeding seasons, where meeting time was a weekend when a pair was first seen together. Shaded areas represent 95% CI of the line. Dots represent the original data points. (Online version in colour.)

## (b) Meeting time and future mating decisions

Model selection on both datasets (2007/2008 to 2009/2010; 2011/2012 to 2013/2014) did not support a direct influence of meeting time on the probability that a pair breeds together again. In the first three years, model selection gave the best support to two models (electronic supplementary material, table S22). One included 'laydate' as the main predictor of divorce of a pair, and the other one 'meeting time' and 'laydate'. However, the credible intervals of the term meeting time (as well as the laydate) overlapped zero. For the second three years, several models gained similar support (electronic supplementary material, table S23). However, models with the term 'meeting time' did not outperform the models without the term (electronic supplementary material, table S23), and the credible intervals of all the parameter estimates in these models overlapped zero.

## 4. Discussion

Our study is, to our knowledge, the first to report that meeting a future partner early translates into breeding success

benefits in a wild population. We found that in all studied years, pairs that met earlier and thus started to socially associate earlier, achieved an earlier laydate, and that these earlier laying pairs had higher breeding success (indirect effect of meeting time via laydate) However, our analysis did not find any evidence of a direct effect of meeting time on the later components of reproductive success (from clutch size to fledging success). Interestingly, our results did not support any direct influence of the time when a pair met (i.e. a potentially longer period of partner assessment) on the future stability of their bond (divorce or fidelity).

The potential effects of meeting and associating with a future breeding partner early have thus far mainly focused on possible hormonal and behavioural synchronization that partners achieve during this period, and were primarily addressed in long-lived species [e.g. 8,16]. In these systems, future pair members sometimes start to associate several years prior to their first breeding, already forming a distinct bond [8,16]. Three main functions of these early pairings have been suggested: direct benefits through higher winter survival of bonded individuals [8], benefits of within pair coordination leading to higher breeding success [19], and

benefits of the prolonged period of partner assessment [17]. Our results are the first to suggest both direct and indirect (i.e. acting via laydate electronic supplementary material, figure S5) links between the time when pair members met for the first time (and started to associate) and their subsequent reproductive success.

Strong selection for an earlier laydate has been detected across great tit populations, including the population studied here (e.g. [33,51,52]). Thus, our finding that meeting a partner earlier in winter leads to an earlier laydate in all of the six seasons studied here implies that there is a selective advantage of earlier pairing in a wild populations and that the 'pair familiarity effects' can extend to the familiarity of partners prior to their first breeding. Our analysis did not support any influence of female or male arrival time on laydate, thus it seems that it is truly the effect of pair meeting time, rather than arrival times, that leads to this pattern. Interestingly, we detected that the relationship between the meeting time of a pair and their laydate was not linear (figure 2). These nonlinear effects were also found in captive bearded reedlings where pairs with the longest pairing period (i.e. 6.5 months) performed significantly better than pairs with a short pairing period (one month), but there was no difference between short and medium (four months) pairing periods [19]. In wild population, it is possible that the influx of new individuals, and the return of some birds that leave the woods during winter, but return to breed [34], maintains the pool of potential partners up to mid-winter (supported by the distribution of arrival times of males and females in our dataset, see the electronic supplementary material). Furthermore, birds of better quality or with a certain behavioural trait(s), which also breed earlier, may pair earlier than birds of lower quality, and that this may, in turn, result in a positive correlation between meeting time and laydate. Going forward, these related concepts could be disentangled using experiments aimed at directly manipulating social associations in the wild and testing the outcomes directly [37,40].

Pair meeting time has not, however, influenced later components of reproductive output in our population directly (but only indirectly via laydate). Thus, our results do not support the hypothesis that familiar pairs might achieve higher behavioural synchronization, and thus make better (more synchronized) parents. Our results also do not directly align with previous findings in great tits [53] that found that baseline corticosterone compatibility increased between two weeks before breeding, and during breeding, and that pairs where this compatibility increased raised more fledglings. However, these previous findings only considered individuals during a short time preceding breeding season, while our study considers a much longer time period.

Interestingly, when considering divorce in subsequent years, we found no evidence that the length of the initial pairing period (i.e. meeting time) influenced future mating decisions (staying with a partner for the next breeding or divorcing a partner) directly. However, it is possible that such effects could arise through affecting the breeding success of the pair (figure 1), as breeding success has previously been shown to predict divorce probability of pairs across monogamous birds, where pairs with lower success are more likely to divorce ([50], also meta-analysis we have conducted, see the electronic supplementary material).

## 5. Conclusion

Our study shows that newly formed pairs generally appear to benefit from beginning to associate earlier in the pre-breeding period, prior to the first breeding together, in a wild population of a monogamous passerine bird. These benefits appear to come in the form of direct positive effects on the pair's laydate, a known predictor of fitness within this species. As such, our study suggests that the benefits of pair familiarity in socially monogamous systems with biparental care might not only arise from repeated breeding with the same partner, but also from a longer time of association prior to the first breeding of a pair. More studies including a range of different species, as well as experimental studies, are needed to explore the generality of this pattern, and the mechanisms underlying it. Such research would further increase our knowledge on the evolution and function of pair bonds and social monogamy.

Ethics. All work was carried out under permission of Oxford University Internal Animal Welfare Committee (Zoology), and all bird ringing and tagging was carried out under standard licencing permissions from the British Trust for Ornithology (BTO) over all of the study years. This particular study did not require taking any individuals into captivity.

Data accessibility. Datasets and code used in the main and electronic supplementary material analysis, including the code to produce the figures are available from the Dryad Digital Repository: https://doi.org/10.5061/dryad.2z34tmpj9 [54].

Authors' contributions. A.C. has conceived the study, conducted the analysis and written the first draft. C.A.H. and J.A.F. have participated in the development of the ideas, and critically reviewed the manuscript. All the authors have participated in data collection.

Competing interests. We declare we have no competing interests.

Funding. This work was supported by the ERC Advanced grant (grant no. AdG 250164) to Ben C. Sheldon, and NWO personal grant (grant no. 016.Veni.181.054) to A.C. J.A.F. was supported by a research fellowship from Merton College and BBSRC (BB/S009752/1) and acknowledges funding from NERC (NE/S010335/1).

Acknowledgements. We are grateful to all the people that helped to collect data on social winter behaviour and breeding data of great tits in Wytham woods. We are especially grateful to B.C. Sheldon who has provided valuable insights and comments about this work through numerous discussions.

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
