## [Reviewer comments · Proceedings of the Royal Society B: Biological Sciences]

Review History

RSPB-2019-2366.R0 (Original submission)

Review form: Reviewer 1

Recommendation

Reject – article is scientifically unsound

Scientific importance: Is the manuscript an original and important contribution to its field?

Acceptable

General interest: Is the paper of sufficient general interest?

Acceptable

Quality of the paper: Is the overall quality of the paper suitable?

Acceptable

Is the length of the paper justified?

Yes

Should the paper be seen by a specialist statistical reviewer?

Yes

Do you have any concerns about statistical analyses in this paper? If so, please specify them explicitly in your report.

Yes

It is a condition of publication that authors make their supporting data, code and materials available - either as supplementary material or hosted in an external repository. Please rate, if applicable, the supporting data on the following criteria.

Is it accessible?

Yes

Is it clear?

Yes

Is it adequate?

Yes

Do you have any ethical concerns with this paper?

No

Comments to the Author

Overall this is a well-presented and well-written ms, with some occasional typo's and grammatical errors. This ms claims to be looking at pre-breeding pair bond formation, and the whole ms is framed in this vein. However, pair association is defined as the first time they were recorded together at the same feeder. The authors acknowledge that this represents a snapshot in time. It is very possible that the pair did not associate after being attracted to the feeder for food. But more worrying for me is that they move as part of a flock in winter, and so occurrence at the feeder records all flock members, not just a pair. This data is therefore recording flock composition, and the two that subsequently bred together may not be anymore associated with each other than they are to any other flock member during the winter season. The authors presumably have the data to determine if pairs were more often in the same flock together throughout winter than other flock members? This is not presented. Overall, I am not convinced that this is a question about pair association.

In addition, I feel that there are areas of the ms that need more clarity to make it clear what was analysed, when, and the biological conclusion that logically follows from a particular analysis.

My detailed comments are as follows:

L 15: this is not true for species that are territorial year-round.

L 34: or because both individuals are 'high quality'

L 41-48: is the initial pair bonding period being referred to here, or is this a cycle that repeats itself each year/breeding season?

L 68 & 69: typos

L 53-58: the problem with this expt was that pairs were formed not through choice by the birds themselves, so likely to have a different outcome from pair formation in the wild

Throughout the intro, it is a bit unclear by what is meant by 'first meet'. Could this be an association of simply a few seconds, typical of individuals that are part of a larger population

moving through the landscape? Or does it mean they associated with each other for a certain period of time that is significantly longer than typical associations with other members of the population? The statement regarding flocks on L 78 relates to this use stated above.

L 72: is meeting considered a pre-breeding mating behaviour? This is ambiguous.

L 76: reference needed

L 93: typo

L 101-111: how big is flock size typically in this spp? This relates to the concern I had above: are these two really associating if they are simply in the same flock together? They are associating with each other no more than with any other member of the flock, which presumably includes many other potential partners?

L 115: do they need to be detected only once to qualify? Also, how is a network different from a flock. The above described a flock. Here, it says network.

L 142" this is a bit vague: a model with a number of both important and unimportant predictors could therefore be chosen as the model with the most support. But how is the reader to know WHICH terms in the model were important? Global models run the risk of including terms that are not good predictors of the data patterns. I assume the author/s chose the right terms to discuss, but would like to see it detailed here, how they determined the terms that are the best predictors.

L147: again, related to the above: unclear why being in the same flock (which is how meeting time is defined) is a good indicator of the start of pair bonding, when there are also other birds in the flock?

L 164: reference for this statement?

L 165-171: it is stated here that previous breeding stages were controlled for, but does not state how they were controlled for. Please specify for clarity

L 175: but likelihood of predation may relate to parental ability? E.g. to build a 'safer' nest, or to defend against predators?

L 175-177: this highlights the problem discussed above: you are applying your assumption to one part of the data, but not the other.

L 183: but several of the GLMs you described above were also not linear.

L 184: typo. Also not performing worse (reference needed for this also) is not a very good justification! It implies that a GLM could have been just as good.

L 185-187: please justify why the smoothing function was needed and how it affected the data.

L 190: suggest you use a term other than 'faithful'. Just as the term 'promiscuous' has been edited out of sexual selection papers, faithful implies something that is not necessarily correct in this context.

L 194: I am not convinced that combining years together where sample size is low is a valid approach, unless you have done an analysis which finds no noticeable difference in trends between these years.

L 209-210: what analysis has been used here?

L 212: remove the word 'clearly'. Also, missing bracket.

L 214: but does this represent pair bonding, or simply flock formation?

L 221: I don't know what this means: 'the best supported model was GAM...' also, it goes on to say it influenced one factor in one year in a non-linear way? This is very confusing.

L 231: unclear how the author/s have come to this conclusion of independent effects: please clarify

L 232: 'depended on winter' - in what way?

L 233 states that fledging success depended on the type of pair, but does not tell us what type of pair had more or less success than the other type of pair.

L 239: missing word

L 241: why is meeting time described as plural here?

L 245: give this statement biological meaning: what does it mean when the slope of meeting time was similar and low in both models?

L 247-9: consider rewriting this: it is confusing to read.

L 255: how can model selection 'give similar and the best support to two models'?

L 257-8: so were there any terms that were good predictors of the data? This is unclear.

L 262: are these two different? From the methods, it seems these two (met for the first time and started to associate) are the same thing?

L 264: is 'correlation' the right word to use here?

L 269: grammar

L 271-272: but if a flocking or group-living spp, this can apply to those individuals that never breed together as well.

L 273: these are not necessarily early pairings though, simply a part of being in the same flock or group

L 296: sci name needed

L 308-310: linkage missing: I am not sure why the results imply this?

L 319-321: references needed for this statement

L 333 but you likely already have this data, as you had tags on many members of the flocks, not just those that paired?

References: these are very heavily weighted toward great tit (and closely related spp) references: suggest a broader range needed.

Table 2: what is the sample size for each analysis? Unclear why the table has P-values, if this is model selection?

Fig 2: graphs and caption both need editing for clarity

Supplemental material: Sample size is needed for each table. Also, it is stated that the overall best model is highlighted in bold, but multiple ones are highlighted in the same table. In subsequent tables: where no models are highlighted, does it mean that no models provided sufficient support for the data patterns? Change Modles to Models

Review form: Reviewer 2

Recommendation

Reject - article is not of sufficient interest (we will consider a transfer to another journal)

Scientific importance: Is the manuscript an original and important contribution to its field?

Good

General interest: Is the paper of sufficient general interest?

Good

Quality of the paper: Is the overall quality of the paper suitable?

Good

Is the length of the paper justified?

Yes

Should the paper be seen by a specialist statistical reviewer?

No

Do you have any concerns about statistical analyses in this paper? If so, please specify them explicitly in your report.

No

It is a condition of publication that authors make their supporting data, code and materials available - either as supplementary material or hosted in an external repository. Please rate, if applicable, the supporting data on the following criteria.

Is it accessible?

Yes

Is it clear?

Yes

Is it adequate?

Yes

Do you have any ethical concerns with this paper?

No

Comments to the Author

The authors use RFID technology to conduct fine-scale tracking and measure associations between breeding pairs in winter, and test for fitness benefits in relation to how early they

interact. This is a well written paper with impressive analyses of good sample sizes, in a well-studied system (great tits at Wytham). I guess what it comes down to is whether this is a novel and significant enough contribution for publication in Proceedings. We already know that such factors as laydate can have a significant impact on fitness in birds. Also, the authors point out the importance of early associations in long-lived species. And they cite an experimental study (captive-based) in which bearded reedlings were allowed to spend different amounts of time together prior to breeding. The primary significance or novelty of this study is that it is the first study in the wild for a monogamous passerine. The study has two aims or questions – first, how does time since first meeting affect a range of breeding components that bear on fitness and second, how this impacts on the future probability of divorce or staying together. The authors focus only on species that have not bred together previously. The focus of this study is on meeting time and unfortunately (perhaps because of difficulty tracking flocks) not any measure of social association prior to breeding. By this, I mean that we don't know anything about the measure or strength of social association prior to this time – how frequently they interact and associate. I imagine this is difficult but if they are visiting feeders regularly, would some index of association in the pre-breeding season be available beyond just when they first meet? It seems like this would really strengthen the study.

The title is informative but quite a mouthful, although I do like the first part. Is it important to mention their mating system? Why not just say “Familiarity breeds success: early pre-breeding meeting has breeding benefits in great tits”?

Pg 1 of intro focuses on the mate familiarity hypothesis, but there is also the reproductive performance hypothesis (alluded to later), which suggests that more experienced parents have a fitness advantage. It's true that this applies less to monogamous species, but monogamous species also divorce. I also found that the intro was almost exclusively focused on birds for quite obvious reasons. Nevertheless, there are papers on other systems alluding to the role of familiarity and its influence on mating. For example, see Leu et al. 2015 Ethology for lizards. I do appreciate that space is a constraint and just suggest a few more sentences to expand beyond birds.

17 delete “to”

21-25 be explicit about the relationship between meeting date and whether a pair stays together or divorces.

68 lower probability

75 I know that great tits are very well studied, but I suggest greatly expanding this paragraph (at least by 2-4 sentences). Given the nature of this study, as a reader, I would like to know some basics about the breeding biology of great tits in the Wytham population. This includes the age they first breed, when do they typically pair up to breed, when do they lay eggs (date range) and is this variable between years in the same pair. Can it depend on a range of environmental variables that can be briefly mentioned? And say something about their mating system. As I understand it from other work, they are socially monogamous but roughly half of broods contain EP offspring? These mating system and life history variables will help the reader place this study in context.

76 short lived (give a range or some other measure, in years)

109-111 I don't understand this. Why are they different (first 3 vs last 3 winters)?

112 Is “preseason bonding” the same as preseason “meeting”? As I understand it you established when unfamiliar pairs first met prior to the breeding season. This might require a reword.

114 Further,

205-206 I would include a sentence explaining why these species were selected. Saying they are closely related is not enough of a justification. I would assume they are similar in their social and reproductive behaviour?

230-231 I didn't get this from the previous sentence?

233 what about the type of pair?

238)

255-256, 257-258 yes, but in what direction? Can you reword to explicitly give the relationship?

266-268 this seems to be in contrast to results?

Decision letter (RSPB-2019-2366.R0)

15-Nov-2019

Dear Dr Culina:

I am writing to inform you that we have now obtained responses from referees on manuscript RSPB-2019-2366 entitled "Familiarity breeds success: earlier winter meeting of a pair relates to breeding benefits in a monogamous passerine population" which you submitted to Proceedings B.

Unfortunately, on the advice of the Associate Editor and the referees, your manuscript has been rejected following full peer review. Competition for space in Proceedings B is currently extremely severe, as many more manuscripts are submitted to us than we have space to print. We are therefore only able to publish those that are exceptional, convincing and present significant advances of broad interest, and must reject many good manuscripts.

On a more positive note, based on the advice we have received, we would like to offer you the opportunity to transfer your manuscript file to another Royal Society journal, Royal Society Open Science. Royal Society Open Science is a fast, open journal publishing high-quality research across all of science and mathematics. The journal operates objective peer review, optional open peer review, and will publish any article deemed to sufficiently advance the field by the reviewers and editors, leaving judgement of potential impact of the work to the reader. The journal publishes Registered Reports and encourages the submission of negative results. You can find out more about the scope of the journal and the benefits of publication here <https://royalsocietypublishing.org/journal/rsos>

If you wish to have your manuscript transferred to Royal Society Open Science please ensure that you revise your text to address all of the reviewers' comments relating to scientific soundness. Please particularly ensure that your conclusions do not overstate the results of your study. Once submitted to Royal Society Open Science your manuscript will be assessed by an Associate Editor who will decide whether further reviewer advice is required. If no further advice is needed and all of your revisions are satisfactory your manuscript will be immediately accepted for publication.

If you agree to transfer your paper, and it is accepted for publication, you will be asked to pay the article processing charge, unless you request a waiver and this is approved by Royal Society Publishing. You can find out more about the charges at <https://royalsocietypublishing.org/rsos/charges>.

You can approve or reject this transfer using the links below:

Approve transfer - *** PLEASE NOTE: This is a two-step process. After clicking on the link, you will be directed to a webpage to confirm. ***

https://mc.manuscriptcentral.com/prsb?URL_MASK=54edd2eacfc9490b89c20c2933dd2c52
After approving the transfer you will need to log in to your Royal Society Open Science author centre (<https://mc.manuscriptcentral.com/rsos>) to complete your the submission. At this stage you will have chance to address any of the reviewers' or editor's concerns.

Reject transfer - *** PLEASE NOTE: This is a two-step process. After clicking on the link, you will be directed to a webpage to confirm. ***

https://mc.manuscriptcentral.com/prsb?URL_MASK=9ffc884035c540f787d0e752a15a0e1f

or by clicking 'approve' or 'reject' in your Author Center.

Once you have approved the transfer you will be prompted to complete the transfer of your article via the Royal Society Open Science submission system.

Please find below the comments received from referees concerning your manuscript, not including confidential reports to the Editor. If you approve transfer to Royal Society Open Science, these reviews will accompany your paper.

Thank you for your interest in Proceedings B.

Sincerely,
Proceedings B
<mailto:proceedingsb@royalsociety.org>

Associate Editor
Board Member: 1
Comments to Author:

This interesting manuscript uses an impressive dataset to investigate the possibility that pair associations well before the breeding season could influence reproductive success later in the season. As the authors note, this is a difficult question to study and there is relatively little known about pair associations for short-lived birds like great tits. However, I agree with both reviewers that there are some serious flaws with the manuscript as it is currently framed and written. As both reviewers note, it is not clear that birds that were recorded in "pair" associations during the winter were actually associating any more closely or more frequently than with other flock members -- the identification of them as a "pair" seems to be made post hoc from their behavior in the breeding season. This suggests to me that there is a potential confound with the time that a given individual shows up in the data set at all (in other words, birds that immigrate into the area relatively late in the winter must, by definition, "pair" later than those that have been present all winter, but there is no evidence that birds that have been present all winter are actually paired earlier). It might be possible to re-do these analyses to investigate whether "pairs" that form earlier are actually pairs, but this distinction must emerge from the data. I think that the data set may be limited in what conclusions can really be drawn about the mechanism behind any kind of hypothesized "familiarity," since direct behavioral observations are largely lacking.

Reviewer(s)' Comments to Author:
Referee: 1

Comments to the Author(s)

Overall this is a well-presented and well-written ms, with some occasional typo's and grammatical errors. This ms claims to be looking at pre-breeding pair bond formation, and the

whole ms is framed in this vein. However, pair association is defined as the first time they were recorded together at the same feeder. The authors acknowledge that this represents a snapshot in time. It is very possible that the pair did not associate after being attracted to the feeder for food. But more worrying for me is that they move as part of a flock in winter, and so occurrence at the feeder records all flock members, not just a pair. This data is therefore recording flock composition, and the two that subsequently bred together may not be anymore associated with each other than they are to any other flock member during the winter season. The authors presumably have the data to determine if pairs were more often in the same flock together throughout winter than other flock members? This is not presented. Overall, I am not convinced that this is a question about pair association.

In addition, I feel that there are areas of the ms that need more clarity to make it clear what was analysed, when, and the biological conclusion that logically follows from a particular analysis.

My detailed comments are as follows:

L 15: this is not true for species that are territorial year-round.

L 34: or because both individuals are 'high quality'

L 41-48: is the initial pair bonding period being referred to here, or is this a cycle that repeats itself each year/breeding season?

L 68 & 69: typos

L 53-58: the problem with this expt was that pairs were formed not through choice by the birds themselves, so likely to have a different outcome from pair formation in the wild

Throughout the intro, it is a bit unclear by what is meant by 'first meet'. Could this be an association of simply a few seconds, typical of individuals that are part of a larger population moving through the landscape? Or does it mean they associated with each other for a certain period of time that is significantly longer than typical associations with other members of the population? The statement regarding flocks on L 78 relates to this use stated above.

L 72: is meeting considered a pre-breeding mating behaviour? This is ambiguous.

L 76: reference needed

L 93: typo

L 101-111: how big is flock size typically in this spp? This relates to the concern I had above: are these two really associating if they are simply in the same flock together? They are associating with each other no more than with any other member of the flock, which presumably includes many other potential partners?

L 115: do they need to be detected only once to qualify? Also, how is a network different from a flock. The above described a flock. Here, it says network.

L 142" this is a bit vague: a model with a number of both important and unimportant predictors could therefore be chosen as the model with the most support. But how is the reader to know WHICH terms in the model were important? Global models run the risk of including terms that are not good predictors of the data patterns. I assume the author/s chose the right terms to discuss, but would like to see it detailed here, how they determined the terms that are the best predictors.

L147: again, related to the above: unclear why being in the same flock (which is how meeting time

is defined) is a good indicator of the start of pair bonding, when there are also other birds in the flock?

L 164: reference for this statement?

L 165-171: it is stated here that previous breeding stages were controlled for, but does not state how they were controlled for. Please specify for clarity

L 175: but likelihood of predation may relate to parental ability? E.g. to build a 'safer' nest, or to defend against predators?

L 175-177: this highlights the problem discussed above: you are applying your assumption to one part of the data, but not the other.

L 183: but several of the GLMs you described above were also not linear.

L 184: typo. Also not performing worse (reference needed for this also) is not a very good justification! It implies that a GLM could have been just as good.

L 185-187: please justify why the smoothing function was needed and how it affected the data.

L 190: suggest you use a term other than 'faithful'. Just as the term 'promiscuous' has been edited out of sexual selection papers, faithful implies something that is not necessarily correct in this context.

L 194: I am not convinced that combining years together where sample size is low is a valid approach, unless you have done an analysis which finds no noticeable difference in trends between these years.

L 209-210: what analysis has been used here?

L 212: remove the word 'clearly'. Also, missing bracket.

L 214: but does this represent pair bonding, or simply flock formation?

L 221: I don't know what this means: 'the best supported model was GAM...' also, it goes on to say it influenced one factor in one year in a non-linear way? This is very confusing.

L 231: unclear how the author/s have come to this conclusion of independent effects: please clarify

L 232: 'depended on winter' - in what way?

L 233 states that fledging success depended on the type of pair, but does not tell us what type of pair had more or less success than the other type of pair.

L 239: missing word

L 241: why is meeting time described as plural here?

L 245: give this statement biological meaning: what does it mean when the slope of meeting time was similar and low in both models?

L 247-9: consider rewriting this: it is confusing to read.

L 255: how can model selection 'give similar and the best support to two models'?

L 257-8: so were there any terms that were good predictors of the data? This is unclear.

L 262: are these two different? From the methods, it seems these two (met for the first time and started to associate) are the same thing?

L 264: is 'correlation' the right word to use here?

L 269: grammar

L 271-272: but if a flocking or group-living spp, this can apply to those individuals that never breed together as well.

L 273: these are not necessarily early pairings though, simply a part of being in the same flock or group

L 296: sci name needed

L 308-310: linkage missing: I am not sure why the results imply this?

L 319-321: references needed for this statement

L 333 but you likely already have this data, as you had tags on many members of the flocks, not just those that paired?

References: these are very heavily weighted toward great tit (and closely related spp) references: suggest a broader range needed.

Table 2: what is the sample size for each analysis? Unclear why the table has P-values, if this is model selection?

Fig 2: graphs and caption both need editing for clarity

Supplemental material: Sample size is needed for each table. Also, it is stated that the overall best model is highlighted in bold, but multiple ones are highlighted in the same table. In subsequent tables: where no models are highlighted, does it mean that no models provided sufficient support for the data patterns? Change Modles to Models

Referee: 2

Comments to the Author(s)

The authors use RFID technology to conduct fine-scale tracking and measure associations between breeding pairs in winter, and test for fitness benefits in relation to how early they interact. This is a well written paper with impressive analyses of good sample sizes, in a well-studied system (great tits at Wytham). I guess what it comes down to is whether this is a novel and significant enough contribution for publication in Proceedings. We already know that such factors as laydate can have a significant impact on fitness in birds. Also, the authors point out the importance of early associations in long-lived species. And they cite an experimental study (captive-based) in which bearded reedlings were allowed to spend different amounts of time together prior to breeding. The primary significance or novelty of this study is that it is the first study in the wild for a monogamous passerine. The study has two aims or questions – first, how does time since first meeting affect a range of breeding components that bear on fitness and second, how this impacts on the future probability of divorce or staying together. The authors focus only on species that have not bred together previously. The focus of this study is on

meeting time and unfortunately (perhaps because of difficulty tracking flocks) not any measure of social association prior to breeding. By this, I mean that we don't know anything about the measure or strength of social association prior to this time – how frequently they interact and associate. I imagine this is difficult but if they are visiting feeders regularly, would some index of association in the pre-breeding season be available beyond just when they first meet? It seems like this would really strengthen the study.

The title is informative but quite a mouthful, although I do like the first part. Is it important to mention their mating system? Why not just say “Familiarity breeds success: early pre-breeding meeting has breeding benefits in great tits”?

Pg 1 of intro focuses on the mate familiarity hypothesis, but there is also the reproductive performance hypothesis (alluded to later), which suggests that more experienced parents have a fitness advantage. It's true that this applies less to monogamous species, but monogamous species also divorce. I also found that the intro was almost exclusively focused on birds for quite obvious reasons. Nevertheless, there are papers on other system alluding to the role of familiarity and it's influence on mating. For example, see Leu et al. 2015 Ethology for lizards. I do appreciate that space is a constraint and just suggest a few more sentences to expand beyond birds.

17 delete “to”

21-25 be explicit about the relationship between meeting date and whether a pair stays together or divorces.

68 lower probability

75 I know that great tits are very well studied, but I suggest greatly expanding this paragraph (at least by 2-4 sentences). Given the nature of this study, as a reader, I would like to know some basics about the breeding biology of great tits in the Wytham population. This includes the age they first breed, when do they typically pair up to breed, when do they lay eggs (date range) and is this variable between years in the same pair. Can it depend on a range of environmental variables that can be briefly mentioned? And say something about their mating system. As I understand it from other work, they are socially monogamous but roughly half of broods contain EP offspring? These mating system and life history variables will help the reader place this study in context.

76 short lived (give a range or some other measure, in years)

109-111 I don't understand this. Why are they different (first 3 vs last 3 winters)?

112 Is “preseason bonding” the same as preseason meeting? As I understand it you established when unfamiliar pairs first met prior to the breeding season. This might require a reword.

114 Further,

205-206 I would include a sentence explaining why these species were selected. Saying they are closely related is not enough of a justification. I would assume they are similar in their social and reproductive behaviour?

230-231 I didn't get this from the previous sentence?

233 what about the type of pair?

238)

255-256, 257-258 yes, but in what direction? Can you reword to explicitly give the relationship?

266-268 this seems to be in contrast to results?

Author's Response to Decision Letter for (RSPB-2019-2366.R0)

See Appendix A.

RSPB-2020-1554.R0

Review form: Reviewer 1

Recommendation

Major revision is needed (please make suggestions in comments)

Scientific importance: Is the manuscript an original and important contribution to its field?

Acceptable

General interest: Is the paper of sufficient general interest?

Good

Quality of the paper: Is the overall quality of the paper suitable?

Good

Is the length of the paper justified?

Yes

Should the paper be seen by a specialist statistical reviewer?

No

Do you have any concerns about statistical analyses in this paper? If so, please specify them explicitly in your report.

Yes

It is a condition of publication that authors make their supporting data, code and materials available - either as supplementary material or hosted in an external repository. Please rate, if applicable, the supporting data on the following criteria.

Is it accessible?

Yes

Is it clear?

Yes

Is it adequate?

Yes

Do you have any ethical concerns with this paper?

No

Comments to the Author

This is my second reading of this ms. I find the clarification about how to differentiate between a pair and just another flock member (and the additional analyses to support this) helpful, thank you. I do however have other areas which I think are still not clear. Also, I found the response to the questions about model selection unclear. On consulting your AIC tables in the supplementary material, I think you have done these as expected for model selection, and so your response, while unclear, seems to not have affected the AIC output. However, it seems some AIC tables are missing? And why are effect sizes and confidence intervals not given for terms in top models? These are important to determine the effect of different predictor terms on data patterns.

L 24: 'opens a new area in the study' – what is the new area? This statement is a bit vague/ambiguous and I suggest deleting it.

L 51: typo 'each-other'

L 55: typo, should read 'reinforcing'

L 69: add reference

L 86: replace 'probably' with 'probability'

L 103: are these annual survival rates?

L 105: are these numbers per nesting attempt? Or per season (multiple nesting attempts?)

L 121: typo: should read 'transponders'

Methods: so data collection is reliant on RFID tagged birds visiting feeders: but how often do birds visit feeders? Do some of them never visit feeders?

L 159: this information is important: the sum of the times when they were observed at all, since it affects how a pair is defined. Feeders give a snapshot in time, and so is a pair defined as being at the same feeder in a weekend for 10 seconds or less? As in, please give the context for 'time observed at all'. How often were they typically observed for per week/month on average? L165 suggests observations include individuals that were detected at a feeder just once, suggesting observations are only a few second long?

L 174: how does this rate (29 and 9) compare to the population average of associations between non-breeding birds for context? As in, is this rate unusual?

L 187: this statement is vague, please clarify 'we compared the estimates of the effects'

IN response to my question about the model selection, the authors have answered as follows: 'The issue here is that it can happen that removing the statistically non significant predictor can also change the estimate of the significant predictor. In our analysis we have opted for a second choice, where we compare plausible model structures, and select the best supported model(s) based on their AIC. These models will indeed contain some insignificant terms, but removing these terms can further influence the estimate of the remaining effects.'

I disagree with this: the removal of a non-significant predictor should not change the estimate of the significant predictor. This might happen if the two terms are correlated, and hence why correlated terms should not be included in the same model together. Model selection involves comparing between different hypotheses. Where two models have similar AICc, the simpler model, with less terms, is typically chosen, since the more complicated model likely has non-significant terms contained within it. Important terms can be determined by determining whether the standardized confidence intervals intersect zero A very helpful guideline for model selection

can be found in this widely cited recent paper Harrison et al 2018: A brief introduction to mixed effects modeling and multi-model inference in ecology.

Methods: GLMMS are described but the use of random terms to account for repeated measures not stated, even though they should be present.

L 268: please put more biology into this. It currently reads as 'supported the effect of meeting month on laydate. This is confusing. Supported what effect? An hypothesized positive or negative effect? Reading this sentence, it is unclear whether meeting month advanced or delayed laydate. This problem is in L 270 as well: significant predictors of hatching and fledgling. OK, they are significant: in what direction?

L 274-275: refer to the table that supports this statement.

L 282-289: this issue of not putting the biology into the results statements affects this paragraph too: terms are stated as important, but directionality of effect is not given.

Fig 3: the interaction in (a) is very difficult to understand. What is the sample size for the four different pair types. The sample size to detect this type of interaction with sufficient statistical power is important.

Top model sets for main results should be present in the main text, not supplemental material. Full model sets are ok for supplemental material, but I feel strongly that the reader should be able to see the top model sets in the main text. It allows the reader to see model support, effect size and directionality, and variance.

Supplemental material: I can only find two AIC model output tables, yet the authors conducted more analyses than this? Where are the other AIC output tables?

Review form: Reviewer 3 (Juan Carlos Senar)

Recommendation

Accept with minor revision (please list in comments)

Scientific importance: Is the manuscript an original and important contribution to its field?

Excellent

General interest: Is the paper of sufficient general interest?

Excellent

Quality of the paper: Is the overall quality of the paper suitable?

Excellent

Is the length of the paper justified?

Yes

Should the paper be seen by a specialist statistical reviewer?

No

Do you have any concerns about statistical analyses in this paper? If so, please specify them explicitly in your report.

No

It is a condition of publication that authors make their supporting data, code and materials available - either as supplementary material or hosted in an external repository. Please rate, if applicable, the supporting data on the following criteria.

Is it accessible?

Yes

Is it clear?

Yes

Is it adequate?

Yes

Do you have any ethical concerns with this paper?

No

Comments to the Author

The authors have done a nice work having into account the different comments by the referees. A critical point was to clearly show that members of a future breeding pair associate together more consistently and generally much more frequently than other flockmates, and I think that authors have provided a solid argument to support their statement. I also appreciate the effort of authors to show that it is meeting time and prolonged association that leads to increase in breeding success.

I also appreciate the effort by the authors “to put (their) study into a wider context of familiarity that (they) had previously focused on”. I really think that the topic of “familiarity” has been for years unappreciated. However, within this wider context I miss some reference to previous significant work on the topic. For instance, Cheetham et al. (2008) nicely reviewed the topic on the importance and advantages of familiarity with potential mates to increase fitness. Two more recent papers in Proceedings have shown that individuals show a marked preference to pair with familiar individuals (i.e. from their social group), which may elicit a so strong attraction as traditional ornaments generally assumed to be used in mate choice (Senar et al. 2013; Thunken et al. 2012). They also discuss on the topic. I think these papers should be used to improve the discussion or set the INT on the important and under-appreciated role of familiarity in mate choice. I think this would give to the paper a wider interest.

In the 70's, Saitou wrote a series of very nice papers on flock dynamics along the year in Great tits, and one of the papers specifically was devoted to pairing dynamics within the flocks (Saito 1979). I think you should have a look to this paper and cite it. Saito found that pairs in Great tits were formed from February on, rather than in autumn, as you found. Previous work on great tits, in addition to Saito (1979), also reported that in spite that individuals could flock together in winter, pairing was really taking place in spring (Hinde 1952; Kluijver 1951). I think this merits some discussion. In medaka fish it has been shown that a close association between the members of the “future” pair allows to increase familiarity between them, which is later basic to increase pairing success (Yokoi et al. 2016), but “pair bond” as such, do not takes place at this period of social association but later. I wonder whether in Great tits it could happen the same. Hence, I also would appreciate some discussion on the topic.

I hope these comments help authors to give their very nice paper a wider interest.

Juan Carlos Senar

References

Cheetham SA, Thom MD, Beynon RJ, Hurst JL (2008) The effect of familiarity on mate choice. *Chemical Signals in Vertebrates* 11:271–280

Hinde RA (1952) The Behaviour of the Great Tit (*Parus major*) and Some Other Related Species. Behaviour. Supplement 2:III-201

Kluijver HN (1951) The population ecology of the great tit, *Parus m. major* L. Ardea 39:1-135

Saito T (1979) Ecological study of social organization in the Great Tit, *Parus major* L. IV. Pair formation and establishment of territory in the members of basic flocks. Misc. Rep. Yamashina Inst. Ornith. 11:172-188

Senar JC, Mateos-González F, Uribe F, Arroyo L (2013) Familiarity adds to attractiveness in matters of siskin mate choice. Proc. R. Soc. B 280:20132361

Thunken T, Meuthen D, Bakker TCM, Baldauf SA (2012) A sex-specific trade-off between mating preferences for genetic compatibility and body size in a cichlid fish with mutual mate choice. Proc. R. Soc. B 279:2959-2964

Yokoi S, Ansai S, Kinoshita M, Naruse K, Kamei Y, Young LJ, Okuyama T, Takeuchi H (2016) Mate-guarding behavior enhances male reproductive success via familiarization with mating partners in medaka fish. Frontiers in Zoology 13:21.

Decision letter (RSPB-2020-1554.R0)

03-Aug-2020

Dear Dr Culina:

Your manuscript has now been peer reviewed and the reviews have been assessed by an Associate Editor. The reviewers' comments (not including confidential comments to the Editor) and the comments from the Associate Editor are included at the end of this email for your reference. As you will see, the reviewers and the Editors have raised some concerns with your manuscript and we would like to invite you to revise your manuscript to address them.

Research ethics:

Use of animals and field studies:

It is a condition of publication that you make available the data and research materials supporting the results in the article (<https://royalsociety.org/journals/authors/author-guidelines/#data>). Datasets should be deposited in an appropriate publicly available repository and details of the associated accession number, link or DOI to the datasets must be included in the Data Accessibility section of the article (<https://royalsociety.org/journals/ethics-policies/data-sharing-mining/>). Reference(s) to datasets should also be included in the reference list of the article with DOIs (where available).

Please submit a copy of your revised paper within three weeks. If we do not hear from you within this time your manuscript will be rejected. If you are unable to meet this deadline please let us know as soon as possible, as we may be able to grant a short extension.

Best wishes,
Dr Sasha Dall
mailto: proceedingsb@royalsociety.org

Associate Editor

Comments to Author:

This re-submitted manuscript was reviewed by one of the previous reviewers (#1) and by one new reviewer (new #2). Both I and reviewer #1 found the manuscript to be substantially improved over the first submission, and were convinced by the authors' argument that their methods adequately measure time since pair-bonding and not simply time since first detection in the same flock (which was a major criticism of the first submission). Reviewer 2 also has an overall positive impression of the paper. However, both reviewers also make a number of useful suggestions for further improvements. Based on these comments and my own reading of the paper, I think that the most significant points can be summarized as follows:

First, the overall effect of pairing date on lay date seems robust. It emerges in both data sets (2008-2010 and 2012-2014) and happens regardless of pair type (experience of the male or female in the pair). However, effects on other components of reproduction, such as hatching and fledging success, get harder to interpret as the data are separated by category. My and reviewer #1's comments on Figure 3 reflect this difficulty: We both question whether the sample sizes in different categories are large enough to reflect meaningful differences. Instances when results differ between the two data sets are perhaps the most challenging to understand, because it is not clear (to me, at least) whether results differ due to the different methods employed in the two periods, biological differences between years (i.e. due to weather), or just random chance. (For example, the statement in the abstract that earlier-meeting pairs had higher fledging success "in 3 of 6 years" is not easy to interpret.)

Second, the results of top models need to be reported in the supplementary tables, as well as in the main text for the most important/significant findings. As reviewer #1 notes, this is necessary to evaluate the significance of predictors that are included in top models. Reviewer #1 also argues for removal of non-significant parameters from top models. While I do not think that this is necessary, I do agree that it is necessary to present the estimates, confidence intervals, and p values for each predictor retained in the top models in order for the reader to evaluate which ones are meaningful. Reviewer #1 also notes that when two competing models have similar AIC scores, the model with fewer terms is typically thought to be more informative. The rule of thumb that I usually follow is that if two models are within 2 AIC units, the model with fewer parameters is more informative (see Arnold, T. W. (2010). Uninformative parameters and model selection using Akaike's Information Criterion. *The Journal of Wildlife Management*, 74(6), 1175-1178.)

Third, the presentation of results in the main text and especially in the supplement needs a lot of work to improve clarity. The captions of most of the supplementary figures are impossible to understand without reference back to the supplementary text and main text, so they add little to the readers' understanding (as currently presented). Figure S8 appears to be identical to Fig 1 in the main text. There are also many typos in the results and captions that add to the overall impression of carelessness. I would recommend going through the results and supplement with a fine-toothed comb to correct these errors and adequately explain the supplementary figures.

Fourth, Reviewer 2 suggests several references to strengthen the introduction, which should be incorporated.

I have made a few line-by-line comments as well, which are below.

20: “experienced higher fledging success in 3 of 6 years”

103: please clarify that this is annual survival (I assume?). How long is average life span in this population? Is the average of 7 chicks fledged for successful nests only, or all nests? Finally, it is probably best to specify that this population is non-migratory and does not undertake local movements during the winter.

107-108: it is unclear whether “most birds recruit into the breeding population in their first year” refers to all one-year-olds. What is the survival rate from hatching to one year, and what proportion of those surviving at 1 year breed in their first year?

110-112: what is the number of nest boxes and breeding pairs in the population? Are the nest boxes protected against predators? I realize that this is an extremely well studied population, but please give basic info for readers so they do not have to refer to other papers.

122-123: what were the sample sizes of marked individuals?

201-203: If I understand this correctly, a simpler way of explaining might be that the first step of model selection was full model subsetting in which candidate models were generated from all combinations of predictors and their two-way interactions.

203-205: Please clarify that this means that meeting time was categorized monthly for the first data set, and weekly for the second data set. This is not really obvious to the reader until later in the results, when it comes as a surprise.

259: This too needs similar clarification -- just a reminder that meeting time was categorized either by month or by week, depending on the data set.

277: Fig 2 - why is the data separated by year in 2a, but not in 2b?

286-289: I think this must be referring to Fig 3b, not 2b. Fledging success is not shown in Fig. 2b. In fact, Fig 2b refers to the later data set, and this paragraph otherwise seems to be discussing the earlier dataset.

Correct the typos in both figure captions.

Overall, results section needs to have the actual results of the top models presented (estimates of each parameter, confidence intervals, and p values)

304: Fig 3 - some of these patterns are quite messy and hard to interpret, particularly 3a. Do these lines reflect anything real, or just odd patterns resulting from small sample sizes? Unless these results have a more compelling explanation that I perceive, I'd be in favor of simplifying the results and eliminating this figure. In 3b, the overall negative effect is obvious (lower fledging success with later pairing) but the differences between categories are more confusing. Why do the most experienced pairs (yellow line) have lower fledging success than the rest? Again, it would be worth considering whether the sample sizes allow for meaningful comparisons between categories. If the power is not sufficiently high, it might be better to pool the data.

Reviewer(s)' Comments to Author:

Referee: 1

Comments to the Author(s).

This is my second reading of this ms. I find the clarification about how to differentiate between a pair and just another flock member (and the additional analyses to support this) helpful, thank you. I do however have other areas which I think are still not clear. Also, I found the response to the questions about model selection unclear. On consulting your AIC tables in the supplementary material, I think you have done these as expected for model selection, and so your response, while unclear, seems to not have affected the AIC output. However, it seems some AIC tables are missing? And why are effect sizes and confidence intervals not given for terms in top models? These are important to determine the effect of different predictor terms on data patterns.

L 24: 'opens a new area in the study' - what is the new area? This statement is a bit vague/ambiguous and I suggest deleting it.

L 51: typo 'each-other'

L 55: typo, should read 'reinforcing'

L 69: add reference

L 86: replace 'probably' with 'probability'

L 103: are these annual survival rates?

L 105: are these numbers per nesting attempt? Or per season (multiple nesting attempts?)

L 121: typo: should read 'transponders'

Methods: so data collection is reliant on RFID tagged birds visiting feeders: but how often do birds visit feeders? Do some of them never visit feeders?

L 159: this information is important: the sum of the times when they were observed at all, since it affects how a pair is defined. Feeders give a snapshot in time, and so is a pair defined as being at the same feeder in a weekend for 10 seconds or less? As in, please give the context for 'time observed at all'. How often were they typically observed for per week/month on average? L165 suggests observations include individuals that were detected at a feeder just once, suggesting observations are only a few second long?

L 174: how does this rate (29 and 9) compare to the population average of associations between non-breeding birds for context? As in, is this rate unusual?

L 187: this statement is vague, please clarify 'we compared the estimates of the effects'

IN response to my question about the model selection, the authors have answered as follows: 'The issue here is that it can happen that removing the statistically non significant predictor can also change the estimate of the significant predictor. In our analysis we have opted for a second choice, where we compare plausible model structures, and select the best supported model(s) based on their AIC. These models will indeed contain some insignificant terms, but removing these terms can further influence the estimate of the remaining effects.'

I disagree with this: the removal of a non-significant predictor should not change the estimate of the significant predictor. This might happen if the two terms are correlated, and hence why correlated terms should not be included in the same model together. Model selection involves comparing between different hypotheses. Where two models have similar AICc, the simpler model, with less terms, is typically chosen, since the more complicated model likely has non-significant terms contained within it. Important terms can be determined by determining whether the standardized confidence intervals intersect zero. A very helpful guideline for model selection can be found in this widely cited recent paper Harrison et al 2018: A brief introduction to mixed effects modeling and multi-model inference in ecology.

Methods: GLMMS are described but the use of random terms to account for repeated measures not stated, even though they should be present.

L 268: please put more biology into this. It currently reads as 'supported the effect of meeting month on laydate. This is confusing. Supported what effect? An hypothesized positive or negative effect? Reading this sentence, it is unclear whether meeting month advanced or delayed laydate. This problem is in L 270 as well: significant predictors of hatching and fledgling. OK, they are significant: in what direction?

L 274-275: refer to the table that supports this statement.

L 282-289: this issue of not putting the biology into the results statements affects this paragraph too: terms are stated as important, but directionality of effect is not given.

Fig 3: the interaction in (a) is very difficult to understand. What is the sample size for the four different pair types. The sample size to detect this type of interaction with sufficient statistical power is important.

Top model sets for main results should be present in the main text, not supplemental material. Full model sets are ok for supplemental material, but I feel strongly that the reader should be able to see the top model sets in the main text. It allows the reader to see model support, effect size and directionality, and variance.

Supplemental material: I can only find two AIC model output tables, yet the authors conducted more analyses than this? Where are the other AIC output tables?

Referee: 3

Comments to the Author(s).

The authors have done a nice work having into account the different comments by the referees. A critical point was to clearly show that members of a future breeding pair associate together more consistently and generally much more frequently than other flockmates, and I think that authors have provided a solid argument to support their statement. I also appreciate the effort of authors to show that it is meeting time and prolonged association that leads to increase in breeding success.

I also appreciate the effort by the authors "to put (their) study into a wider context of familiarity that (they) had previously focused on". I really think that the topic of "familiarity" has been for years unappreciated. However, within this wider context I miss some reference to previous significant work on the topic. For instance, Cheetham et al. (2008) nicely reviewed the topic on the importance and advantages of familiarity with potential mates to increase fitness. Two more recent papers in Proceedings have shown that individuals show a marked preference to pair with familiar individuals (i.e. from their social group), which may elicit a so strong attraction as traditional ornaments generally assumed to be used in mate choice (Senar et al. 2013; Thunken et al. 2012). They also discuss on the topic. I think these papers should be used to improve the discussion or set the INT on the important and under-appreciated role of familiarity in mate choice. I think this would give to the paper a wider interest.

In the 70's, Saitou wrote a series of very nice papers on flock dynamics along the year in Great tits, and one of the papers specifically was devoted to pairing dynamics within the flocks (Saito 1979). I think you should have a look to this paper and cite it. Saito found that pairs in Great tits were formed from February on, rather than in autumn, as you found. Previous work on great tits, in addition to Saito (1979), also reported that in spite that individuals could flock together in winter, pairing was really taking place in spring (Hinde 1952; Kluijver 1951). I think this merits some discussion. In medaka fish it has been shown that a close association between the members of the "future" pair allows to increase familiarity between them, which is later basic to increase pairing success (Yokoi et al. 2016), but "pair bond" as such, do not takes place at this period of social association but later. I wonder whether in Great tits it could happen the same. Hence, I also would appreciate some discussion on the topic.

I hope these comments help authors to give their very nice paper a wider interest.
Juan Carlos Senar

References

Cheetham SA, Thom MD, Beynon RJ, Hurst JL (2008) The effect of familiarity on mate choice. *Chemical Signals in Vertebrates* 11:271-280

Hinde RA (1952) The Behaviour of the Great Tit (*Parus major*) and Some Other Related Species. *Behaviour. Supplement* 2:III-201

Kluijver HN (1951) The population ecology of the great tit, *Parus m. major* L. *Ardea* 39:1–135

Saito T (1979) Ecological study of social organization in the Great Tit, *Parus major* L. IV. Pair formation and establishment of territory in the members of basic flocks. *Misc. Rep. Yamashina Inst. Ornith.* 11:172–188

Senar JC, Mateos-González F, Uribe F, Arroyo L (2013) Familiarity adds to attractiveness in matters of siskin mate choice. *Proc. R. Soc. B* 280:20132361

Thunken T, Meuthen D, Bakker TCM, Baldauf SA (2012) A sex-specific trade-off between mating preferences for genetic compatibility and body size in a cichlid fish with mutual mate choice. *Proc. R. Soc. B* 279:2959–2964

Yokoi S, Ansai S, Kinoshita M, Naruse K, Kamei Y, Young LJ, Okuyama T, Takeuchi H (2016) Mate-guarding behavior enhances male reproductive success via familiarization with mating partners in medaka fish. *Frontiers in Zoology* 13:21.

Author's Response to Decision Letter for (RSPB-2020-1554.R0)

See Appendix B.

RSPB-2020-1554.R1 (Revision)

Review form: Reviewer 3 (Juan Carlos Senar)

Recommendation

Accept as is

Scientific importance: Is the manuscript an original and important contribution to its field?

Excellent

General interest: Is the paper of sufficient general interest?

Excellent

Quality of the paper: Is the overall quality of the paper suitable?

Excellent

Is the length of the paper justified?

Yes

Should the paper be seen by a specialist statistical reviewer?

No

Do you have any concerns about statistical analyses in this paper? If so, please specify them explicitly in your report.

No

It is a condition of publication that authors make their supporting data, code and materials available - either as supplementary material or hosted in an external repository. Please rate, if applicable, the supporting data on the following criteria.

Is it accessible?

Yes

Is it clear?

Yes

Is it adequate?

Yes

Do you have any ethical concerns with this paper?

No

Comments to the Author

The authors have done a good job reviewing the paper.

Decision letter (RSPB-2020-1554.R1)

06-Nov-2020

Dear Dr Culina:

Your manuscript has now been peer reviewed and the reviews have been assessed by an Associate Editor. The reviewers' comments (not including confidential comments to the Editor) and the comments from the Associate Editor are included at the end of this email for your reference. As you will see, the reviewers and the Editors have raised some concerns with your manuscript and we would like to invite you to revise your manuscript to address them.

Research ethics:

Use of animals and field studies:

It is a condition of publication that you make available the data and research materials supporting the results in the article (<https://royalsociety.org/journals/authors/author-guidelines/#data>). Datasets should be deposited in an appropriate publicly available repository and details of the associated accession number, link or DOI to the datasets must be included in the Data Accessibility section of the article (<https://royalsociety.org/journals/ethics-policies/data-sharing-mining/>). Reference(s) to datasets should also be included in the reference list of the article with DOIs (where available).

Please submit a copy of your revised paper within three weeks. If we do not hear from you within this time your manuscript will be rejected. If you are unable to meet this deadline please let us know as soon as possible, as we may be able to grant a short extension.

Best wishes,
 Dr Sasha Dall
 Editor, Proceedings B
 mailto: proceedingsb@royalsociety.org

Associate Editor
 Board Member: 1
 Comments to Author:

This well-written and well-presented paper on the fitness consequences of meeting time in pairs of great tits is a revision of an earlier submission. The authors have re-done the statistical analysis as recommended, which has changed the results somewhat. The authors now find that pairs that meet earlier in the winter experience increased reproductive success in the breeding season, but this is due to the effect of lay date. Pairs that meet earlier tend to lay earlier and to have larger clutches, which leads to a higher number of offspring fledged.

Although the revision is much improved (and the authors have supplied much more information on the statistics in supplementary tables, which is great), I still have two remaining concerns. First, as currently written, the abstract implies that pairing time affects lay date but does not affect subsequent reproductive success. This makes the title and conclusions seem unsupported. After reading the ms, I realized that pairing time actually does affect reproductive output, but it does it indirectly through the effect on lay date (in other words, when controlling for lay date, meeting time does not affect rs). This is important because it means that there is no additional advantage to a longer pre-breeding period with a social mate (for example, increased pair coordination). The introduction similarly fails to make it clear that it is already known that earlier lay date leads to increased reproductive output (this needs to be explained and earlier studies cited), and the results mention it only superficially. Given that this is a crucial link in the argument that meeting time has subsequent fitness effects, the authors need to explain this more clearly and not assume that the readers are already familiar with the previous work. The below line-by-line comments highlight areas where this information is missing in the ms.

Second, the central finding of the paper is that earlier meeting time leads to earlier lay date, illustrated in Fig. 2. However, it is not clear to me why meeting date is not standardized within years the way that lay date is. It is difficult to interpret a figure that shows a calendar date on the x axis and a standardized date on the y axis. Of course, I understand why lay date is standardized -- because in some years laying is earlier than in other years, and you are interested in the timing of laying of an individual pair relative to the rest of the population in that year. But it seems like the same logic should apply for meeting time: you are not necessarily interested in the calendar date, but in whether the pair met earlier or later in the year than other pairs in the population.

I thank the authors for their changes to the statistical analysis and would welcome a revision if the two above points can be addressed.

Line-by-line comments:

19-21: This needs to be clarified. If I understand the results correctly, meeting time influences reproductive output indirectly, because pairs that meet earlier also breed earlier, and pairs that breed earlier have larger clutches

76-78: This needs to be expanded to explain that it is already known that earlier laying leads to fitness benefits, and to explain what those benefits are and cite the studies that have found them. Essentially you are testing three hypotheses: 1) that earlier pairing does not change subsequent reproductive timing or outcomes (null); 2) that earlier pairing leads to reproductive benefits directly, via increased coordination or improved parental care; or 3) that earlier pairing leads to reproductive benefits indirectly, via earlier laying.

279-281: But did lay date itself affect reproductive success (clutch size and hatching and fledging success)? It's still not clear to me whether earlier laying translated into reproductive benefits, or whether other studies have found this pattern but it was not actually recapitulated here.

300-304: This section seems to say that lay date did have knock-on effects of reproductive output, but there are no actual results here. Table S18 shows that the best-supported model for clutch size included lay date as the sole predictor.

336: This needs to be explained in the introduction.

376: the phrasing here of "direct positive benefits" is a bit confusing, given that you have described the effect of meeting time on reproductive success (mediated by lay date) as an indirect effect in Fig. 1 and the introduction. It would be clearer to say that meeting time influences lay date, which in turn has positive effects on reproduction.

Reviewer(s)' Comments to Author:

Referee: 3

Comments to the Author(s)

The authors have done a good job reviewing the paper.

Author's Response to Decision Letter for (RSPB-2020-1554.R1)

See Appendix C.

Decision letter (RSPB-2020-1554.R2)

27-Nov-2020

Dear Dr Culina

I am pleased to inform you that your manuscript entitled "Familiarity breeds success: pairs that meet earlier experience increased breeding performance in a wild bird population" has been accepted for publication in Proceedings B.

Open Access

Paper charges

Sincerely,

Dr Sasha Dall
Editor, Proceedings B
mailto: proceedingsb@royalsociety.org

Associate Editor:

Comments to Author:

I thank the authors for their careful revisions, which address my previous concerns. I am looking forward to seeing this exciting manuscript in publication!

Appendix A

Associate Editor

Board Member: 1

Comments to Author:

This interesting manuscript uses an impressive dataset to investigate the possibility that pair associations well before the breeding season could influence reproductive success later in the season. As the authors note, this is a difficult question to study and there is relatively little known about pair associations for short-lived birds like great tits. However, I agree with both reviewers that there are some serious flaws with the manuscript as it is currently framed and written. As both reviewers note, it is not clear that birds that were recorded in "pair" associations during the winter were actually associating any more closely or more frequently than with other flock members -- the identification of them as a "pair" seems to be made post hoc from their behavior in the breeding season. This suggests to me that there is a potential confound with the time that a given individual shows up in the data set at all (in other words, birds that immigrate into the area relatively late in the winter must, by definition, "pair" later than those that have been present all winter, but there is no evidence that birds that have been present all winter are actually paired earlier). It might be possible to re-do these analyses to investigate whether "pairs" that form earlier are actually pairs, but this distinction must emerge from the data. I think that the data set may be limited in what conclusions can really be drawn about the mechanism behind any kind of hypothesized "familiarity," since direct behavioral observations are largely lacking.

RESPONSE: We are grateful of the largely positive comments from the reviewers and the associate editor. These comments have identified a lack of clarity regarding an important point in this manuscript, that "it is not clear that birds that were recorded in 'pair' associations during the winter were actually associating any more closely or more frequently than with other flock members". Importantly, this is a misunderstanding, and – while we accept responsibility for this misunderstanding, we have, based on all the comments, addressed it in full. We have thoroughly revised the manuscript to ensure:

- 1) We clearly state that we know, based on the very robust analyses in prior work on the same population and even the same dataset, that a pair, once they meet, spend time in the same flock. They appear in the same flock more frequently and more consistency than any other two birds with a similar foraging pattern. Further, pair associations throughout the winter are stronger than those to other flock members, and the meeting time (as we define it in this contribution) reflects the start of pair bonding. This is exactly why we have chosen this very system to address the question on meeting time and breeding success. We have also added some additional analysis to this manuscript to show that our results are robust to differences in male and female arrival times in the population, their gregariousness during the meeting month/weekend, as well as to

the addition of several different measures that express the strength of their winter bond. By necessity, these pairs are identified ‘post-hoc’, as we consider all ‘pairs’ observed breeding together in the spring.

- 2) We clearly discuss two type of familiarity with a partner, and describe where our study fits within these. It has been widely accepted that familiarity with another individual does not have to arise from repeated associations with that individual – e.g. we now refer to the work on rodents that shows that familiarity can arise via short-term encounters, based sometimes only on olfactory cues, and that this kind of short-term familiarity can influence later mate choice. For species with bi-parental care (like most of the birds), another type of familiarity, that of repeated association with the future partner, is interesting to study as in this way paired individuals can indeed develop compatible behaviours, or achieve hormonal similarity. As we discuss in the first point, previous analysis has extensively demonstrated that breeding pairs in our system do spend time together after they first meet and that the second type of familiarity is expected in our system. We did not originally go into discussion regarding this in the manuscript (because we know that for our system, meeting time is a good proxy for repeated association of pair members), but our revision includes this discussion, which we also find as helpful to put our study into a wider context of familiarity that we have previously focused on. Finally, the initial meeting time is actually a very objective and intuitive way of measuring ‘familiarity’, unlike general ‘association’. Unlike ‘first meeting time’, there is no clear definition of general ‘association’. Indeed, association could, for instance, be defined as total amount of time together, or time together recently, or time together in the past, or relative time together compared to with others etc. On the other hand, ‘first meeting time’ only has one measure. As such, we believe this is an advantage here (as a single clear measure is more objective than choosing between multiple different potential measures). Nevertheless, we now also consider these alternative measures of social association/familiarity in our additional supplementary analysis, and show that the same conclusions still hold.
- 3) We avoid the use of a term ‘pair-bonded’ within the text when referring to pre-breeding associations. While we are quite certain that the winter bonds of females and males we observe in our system can be called pair bonds through in winter, we also agree that there is not straight-forward way to exactly define a pair-bond in this sense,

and what represents a true pair bond. This is particularly important as a mating pair is commonly defined on the basis of breeding event (as we use here too). Thus, we opted to use a more neutral language to only state that birds do associate from the meeting time on, and that they do so more frequently than any other birds. For the purpose of this study it is only important that the birds spend time together and thus have the opportunity to develop compatible behaviours and physiologically adjust to each other.

Referee: 1

Comments to the Author(s)

Overall this is a well-presented and well-written ms, with some occasional typo's and grammatical errors. This ms claims to be looking at pre-breeding pair bond formation, and the whole ms is framed in this vein. However, pair association is defined as the first time they were recorded together at the same feeder. The authors acknowledge that this represents a snapshot in time. It is very possible that the pair did not associate after being attracted to the feeder for food. But more worrying for me is that they move as part of a flock in winter, and so occurrence at the feeder records all flock members, not just a pair. This data is therefore recording flock composition, and the two that subsequently bred together may not be anymore associated with each other than they are to any other flock member during the winter season. The authors presumably have the data to determine if pairs were more often in the same flock together throughout winter than other flock members? This is not presented. Overall, I am not convinced that this is a question about pair association. In addition, I feel that there are areas of the ms that need more clarity to make it clear what was analysed, when, and the biological conclusion that logically follows from a particular analysis.

RESPONSE: We thank the reviewer for positive comments, and for several important points that he/she has highlighted and which we used to improve our contribution. As we describe in detail in our response to the associate editor, we have a very solid and robust evidence that in our system (based on the same data) from the previous work the period when a pair was first seen to be associated is a good approximation for pair bonding (see response 1). This is exactly one of the reasons why we have decided to address the question of familiarity with a partner and breeding success using this system. However, we fully agree that we were not clear about this in the first version of this manuscript, and we have now thoroughly revised our introduction and methods (including a large amount of additional analysis on the robustness of our conclusions, and 6 additional supplementary figures) to address this issue. Further, as we discuss in our response 3 to the associate editor, we have opted not to imply

that the pair bond started when the pair first met (although we strongly believe we could say so, it is still subjective if this qualifies as pair bond rather than just a very strong bond) and opted to just use a very objective wording that this represents the time when a strong and consistent association between these individuals generally begins.

We have also revised our method section to make it easier to follow our exact procedures, and we believe that in this way we have also made the link between our results and conclusions clearer.

COMMENT: L 15: this is not true for species that are territorial year-round.

RESPONSE: We are unsure whether this remark concerns the statement that pairs often meet and spend time together before they begin reproduction, or the statement about difficulties in studying this period. We believe this regards the later. We have toned down this statement by replacing '*notoriously difficult*' with '*challenging*'

COMMENT: L 34: or because both individuals are 'high quality'

RESPONSE: We have added several references to make it clear that there are several factors that make it difficult (but not impossible) to study the effect of the pair association duration itself on the breeding success. For example, these are quality, age and experience-related increases in reproduction.

COMMENT L 41-48: is the initial pair bonding period being referred to here, or is this a cycle that repeats itself each year/breeding season?

RESPONSE: It is initial pair bonding period. We have ensured this is clear. For an individual this period can repeat every year if the individual breeds with a new partner. Specifically, we have now rewritten this sentence to read as '*This opens up an interesting possibility that being familiar with a future breeding partner before the first breeding event*'. In line with what we discuss in the reply to the associate editor (point 3), we have now also avoided using the subjective term 'pair bonding', and rather use term 'pair association'.

COMMENT L 68 & 69: typos

RESPONSE: Amended. Thank you.

COMMENT L 53-58: the problem with this expt was that pairs were formed not through choice by the birds themselves, so likely to have a different outcome from pair formation in the wild

RESPONSE: We have added *randomly matched* to make this clear.

COMMENT: Throughout the intro, it is a bit unclear by what is meant by ‘first meet’. Could this be an association of simply a few seconds, typical of individuals that are part of a larger population moving through the landscape? Or does it mean they associated with each other for a certain period of time that is significantly longer than typical associations with other members of the population? The statement regarding flocks on L 78 relates to this use stated above.

RESPONSE: In our specific system, as we now describe in detail in the method section, and also mention in the introduction, birds needed to be detected as a part of the same foraging flocking event at least once in a given month/weekend to be considered to meet. Given that the typical flock size experienced by an individual in this population is 5 (Farine et al. Roy Soc Open Sci), we assume that birds indeed had the opportunity to meet one another. However, the meeting time in our analysis is defined as the first month/weekend in which a pair was observed in the same flock, and in this month/weekend they were usually observed more than once in the same flock. While we already knew (based on the previous work on the same population) that birds of a new pair really do associate more often and more consistently than any other two birds with a similar foraging pattern, and we agree that this was unclear from the previous version of the manuscript. Thus, in the revised manuscript we now justify our decision based on the previous work, as well as adding extensive additional analysis to the manuscript. We base these analyses on the Simple Ratio Index (which represents the affinity of two individuals that repeatedly appear in the same flocks rather than in separate flocks) in the meeting month and show that the bond SRI is higher than the 75% of the distribution of all SRIs of all birds for that month. Further, for the majority of males and females their future breeding partner among their 25% strongest associates in the meeting month/weekend. Please see the new supplementary material for more details.

Further, in a general context of familiarity, we now add an additional section to the introduction to describe two main types of familiarity – that based on short-term encounters, and that based on repeated association with the same individual (please see response to the associate editor, point 2).

COMMENT: L 72: is meeting considered a pre-breeding mating behaviour? This is ambiguous.

RESPONSE: We agree and have excluded this statement.

COMMENT L 76: reference needed

RESPONSE: Added Gosler 1993

COMMENT L 93: typo

RESPONSE: Corrected

COMMENT: L 101-111: how big is flock size typically in this spp? This relates to the concern I had above: are these two really associating if they are simply in the same flock together? They are associating with each other no more than with any other member of the flock, which presumably includes many other potential partners?

RESPONSE: In our population, the flock size experienced by an average individual (typical flock size) is around five birds. We now add this information to the manuscript (lines 155/116). These individuals do associate (please see the response to the previous comment). Based on this comment we have also rerun our models while controlling for individual gregariousness (number of associates) in the meeting month/weekend to account for the possibility that individuals which hold more associates are more likely to be classified as meeting their future partner sooner.

COMMENT: L 115: do they need to be detected only once to qualify? Also, how is a network different from a flock. The above described a flock. Here, it says network.

RESPONSE: We agree that we did not clearly describe the difference between the flock and the social network, and we now clearly outline this in the Methods (section *Meeting time and social structure*). In short, a flock is a group of individuals foraging together. When this group comes to a feeder, we detect the individuals in the flock, termed as a flocking event (based on a machine learning algorithm). Based on these flocking events we calculate social network for a certain time period (e.g. month, weekend, or all winter), where individuals represent network nodes, and SRI (Simple Ration Index) represents the edge between two individuals (0 represents no association, 1 represents that they were never seen apart).

COMMENT: L 142” this is a bit vague: a model with a number of both important and unimportant predictors could therefore be chosen as the model with the most support. But how is the reader to know WHICH terms in the model were important? Global models run the risk of including terms that are not good predictors of the data patterns. I assume the author/s chose the right terms to discuss, but would like to see it detailed here, how they determined the terms that are the best predictors.

RESPONSE: This is correct. Currently there is no consensus on whether backward models selection is better than model selection based on comparison of the performance of the candidate models. The issue here is that it can happen that removing the statistically non significant predictor can also change the estimate of the significant predictor. In our analysis we have opted for a second choice, where we compare plausible model structures, and select the best supported model(s) based on their AIC. These models will indeed contain some insignificant terms, but removing these terms can further influence the estimate of the remaining effects. We explain our choice of the explanatory variables in the methods.

COMMENT: L147: again, related to the above: unclear why being in the same flock (which is how meeting time is defined) is a good indicator of the start of pair bonding, when there are also other birds in the flock?

RESPONSE: We explain this in our previous comments. Further, to avoid subjectivity in deciding what qualifies as a pair bond, rather than just a bond, we use term ‘when a pair started to associate’ through the text.

COMMENT: L 164: reference for this statement?

RESPONSE: There is no reference, it is just the property of our dataset. We add 'in our dataset' to make this clear.

COMMENT: L 165-171: it is stated here that previous breeding stages were controlled for, but does not state how they were controlled for. Please specify for clarity

RESPONSE: We have now revised this entire section, which also includes addressing this comment. We further submit our analysis code where the exact structure of all models can be seen.

COMMENT: L 175: but likelihood of predation may relate to parental ability? E.g. to build a 'safer' nest, or to defend against predators?

RESPONSE: This is true, and because we do not know if the likelihood of a complete failure depends on parental ability, or random event (or both), we have analysed both, the data including fledgling failures, and those excluding the failures. We added the sample size to the text to further aid to the understanding of our procedures.

COMMENT: L 175-177: this highlights the problem discussed above: you are applying your assumption to one part of the data, but not the other.

RESPONSE: We are unsure about what the reviewer is referring to precisely here. In the analysis on fledging success, we conduct the additional analysis on only those pairs that have fledged at least one chick (to account for the possibility that complete failure might reflect a chance event rather than parental ability). However, this does not mean that we assume that that failure reflects a chance event (but rather that we do not know how much it is a chance event, and how much does it reflect parental ability, that's why two analysis). As for hatching failure, we do not do this additional analysis because there is only one pair that experiences hatching failure in all of the 6 years.

COMMENT: L 183: but several of the GLMs you described above were also not linear.

RESPONSE: We have now revised our methods section, which also includes removing this statement.

COMMENT: L 184: typo. Also not performing worse (reference needed for this also) is not a very good justification! It implies that a GLM could have been just as good.

RESPONSE: After reading the comments, and consulting some additional literature, we have decided to use polynomial glm, rather than GAM.

COMMENT: L 185-187: please justify why the smoothing function was needed and how it affected the data.

RESPONSE: Please see the above response.

COMMENT: L 190: suggest you use a term other than ‘faithful’. Just as the term ‘promiscuous’ has been edited out of sexual selection papers, faithful implies something that is not necessarily correct in this context.

RESPONSE: While we do agree that the term might be slightly anthropomorphic, it is a term that has established itself in the literature on pair bond stability and has been widely used. We define faithful as breeding with the same social partner in both breeding season, which is clear from the text. However, we decided to use this term (and the term divorce) as infrequently as possible, and rather describe the process (i.e. pairs that do/do not stay to breed together to the next breeding event).

COMMENT: L 194: I am not convinced that combining years together where sample size is low is a valid approach, unless you have done an analysis which finds no noticeable difference in trends between these years.

RESPONSE: We have not combined all of the years in the same analysis. Rather we combine the first three, and the last three years. This pooling of years was only done for the analysis on influence of meeting time on later divorce, as it is impossible to consider each year separately due to a very low sample sizes (please see table 2 for a sample size). Sample size in the first study period was only 17 pairs (and only 2 pairs for the last winter in this dataset). Thus, we did not include the effect of a year in this analysis. However, as the sample size was higher for the second study period (50 pairs), based on the reviewers comment, we rerun the analysis to control for ‘winter’ effect for this study period. However, models with ‘winter’ preformed less well than the models without this term, and our results remained unchanged. We hope that the manuscript is clear about our methods here.

COMMENT: L 209-210: what analysis has been used here?

RESPONSE: It is a meta-analysis. We add ‘meta’ before the ‘analysis’ to make this clear. The details on the meta-analysis are provided in the supplement.

COMMENT: L 212: remove the word ‘clearly’. Also, missing bracket.

RESPONSE: Removed and brackets added

COMMENT: L 214: but does this represent pair bonding, or simply flock formation?

RESPONSE: Please see our response to the several previous comments. This likely represents pair bonding, but it definitely represses the start of repeated and frequent association with a partner. We are now clear about this through the text.

COMMENT: L 221: I don’t know what this means: ‘the best supported model was GAM...’ also, it goes on to say it influenced one factor in one year in a non-linear way? This is very confusing.

RESPONSE: We have substantially revised the Results section, and we believe this comments has been addressed. Further, based on the comments, and after reading some additional literature, we have opted against using GAM models.

COMMENT: L 231: unclear how the author/s have come to this conclusion of independent effects: please clarify

RESPONSE: We have substantially revised the Results section, and we believe this comment has been addressed. In short, if meeting time is significant in the model that also controls for the laydate (in models for hatching and fledging success), that means that meeting time still has an effect other than its effect via laydate. We know that meeting time affects laydate, and if laydate affects fledging success this means that meeting time acts on fledging success both via laydate, but also over-and-above this as well (and, for instance, likely as of the increased coordination of more familiar pairs)

COMMENT: L 232: ‘depended on winter’ – in what way?

RESPONSE: We have substantially revised the Results section, and addressed this comment.

COMMENT: L 233 states that fledging success depended on the type of pair, but does not tell us what type of pair had more or less success than the other type of pair.

RESPONSE: We have substantially revised the Results section, and we addressed this comment. Also, Figure 2b shows the estimated mean fledging success for different pair types in relation to meeting time

COMMENT: L 239: missing word

RESPONSE: Missing word (dataset) added

COMMENT: L 241: why is meeting time described as plural here?

RESPONSE: That was a mistake. Changed to a singular now

COMMENT: L 245: give this statement biological meaning: what does it mean when the slope of meeting time was similar and low in both models?

RESPONSE: This means that the estimates of the slope of the regression line quantifying the influence of meeting time on laydate were similar. However, we have substantially revised the Results section and within this removed this sentence.

COMMENT: L 247-9: consider rewriting this: it is confusing to read.

RESPONSE: We have substantially revised the Results section, and the revisions apply to this sentence too.

COMMENT: L 255: how can model selection ‘give similar and the best support to two models’?

RESPONSE: this means that the AIC of the models were within 2 AIC difference, thus they have gained similar support. We have substantially revised the Results section and the revisions apply to this sentence too.

COMMENT: L 257-8: so were there any terms that were good predictors of the data? This is unclear.

RESPONSE: We have substantially revised the Results section. We have now clearly stated that for several of the breeding success components, only the preceding component of fitness (e.g. clutch size for number of hatchlings) or laydate (for e.g. for hatching and fledging success in the second set of years) were significant predictors.

COMMENT: L 262: are these two different? From the methods, it seems these two (met for the first time and started to associate) are the same thing?

RESPONSE: The wording has been revised in line with the previous comments. In our study system birds do start to associate when they first meet.

COMMENT: L 264: is 'correlation' the right word to use here?

RESPONSE: We change 'correlation' into 'effect'

COMMENT: L 269: grammar

RESPONSE: Corrected

COMMENT: L 271-272: but if a flocking or group-living spp, this can apply to those individuals that never breed together as well.

RESPONSE: We are not sure this comment refers to exactly. Individuals that never breed together can also meet in the non breeding period (and associate). However, these associations were not the primary interest in our manuscript. Further, in our system we know that these associations are weaker than the association between future breeding pair members. We now demonstrate this extensively in our supplementary material.

COMMENT: L 273: these are not necessarily early pairings though, simply a part of being in the same flock or group

RESPONSE: The discussion section has been thoroughly revised, and we made sure not to use the term early pairings as this term brings some subjectivity. We know in our system (as discussed in several prior replies) birds of a future breeding pair do start to associate, and do so more frequently and more consistently than other birds, in the month/weekend when they have first met. However, in this specific case the reviewer refers to, we just use the same term as the literature we refer to.

COMMENT: L 296: sci name needed

RESPONSE: We have already stated scientific name when the species was first mentioned in the text (line 52)

COMMENT: L 308-310: linkage missing: I am not sure why the results imply this?

RESPONSE: We have rewritten this to read: *'Higher fledging success of pairs that met earlier compared to pairs that met later implies that these pairs are better able to raise their chicks. While it is difficult to directly examine the mechanism behind this relationship, it does support the hypothesis that cooperation and coordination between pair members increases*

not only with repeated breeding of a pair [1], but during association in the non-breeding period. A previous study on great tits [48] found that baseline corticosterone compatibility increased between two weeks before breeding season, and breeding season, and that pairs where this compatibility increased raised more fledglings.'

COMMENT: L 319-321: references needed for this statement

RESPONSE: This sentence was removed during the restructuring of the Discussion.

COMMENT: L 333 but you likely already have this data, as you had tags on many members of the flocks, not just those that paired?

RESPONSE: Yes, we have data that may potentially be useful for addressing this, but this would require a different/new framework than used here, and was outside of the focus of our manuscript, but we agree that it would be a very promising future consideration.

COMMENT: References: these are very heavily weighted toward great tit (and closely related spp) references: suggest a broader range needed.

RESPONSE: We have added a broader spectrum of references, which includes monogamous fish, rodents, and lagomorphs. However, most of the studies on the topic of wild pair bonding have been conducted on birds, as a model group to study these topics.

COMMENT: Table 2: what is the sample size for each analysis? Unclear why the table has P-values, if this is model selection?

RESPONSE: We have now replaced this table with the table that describes our methodological approach to aid better understanding of our methods and thus of the derived results (also this now becomes Table 1). Further, all of our data and code will be publicly available at Zenodo repository, and thus all the results with the exact values of the model estimates, can be repeated.

COMMENT: Fig 2: graphs and caption both need editing for clarity

RESPONSE: We have edited this figure and captions

COMMENT: Supplemental material: Sample size is needed for each table. Also, it is stated that the overall best model is highlighted in bold, but multiple ones are highlighted in the same table. In subsequent tables: where no models are highlighted, does it mean that no models provided sufficient support for the data patterns? Change Modles to Models

RESPONSE: We have revised the Supplementary material and added the sample size as suggested. We removed the highlighting as the reader can conclude what the best model is based on the AIC.

We thank Reviewer 1 again for their encouraging comments and helpful, and extensive, suggestions.

Referee: 2

Comments to the Author(s)

The authors use RFID technology to conduct fine-scale tracking and measure associations between breeding pairs in winter, and test for fitness benefits in relation to how early they interact. This is a well written paper with impressive analyses of good sample sizes, in a well-studied system (great tits at Wytham). I guess what it comes down to is whether this is a novel and significant enough contribution for publication in Proceedings. We already know that such factors as laydate can have a significant impact on fitness in birds. Also, the authors point out the importance of early associations in long-lived species. And they cite an experimental study (captive-based) in which bearded reedlings were allowed to spend different amounts of time together prior to breeding. The primary significance or novelty of this study is that it is the first study in the wild for a monogamous passerine. The study has two aims or questions—first, how does time since first meeting affect a range of breeding components that bear on fitness and second, how this impacts on the future probability of divorce or staying together. The authors focus only on species that have not bred together previously. The focus of this study is on meeting time and unfortunately (perhaps because of difficulty tracking flocks) not any measure of social association prior to breeding. By this, I mean that we don't know anything about the measure or strength of social association prior to this time—how frequently they interact and associate. I imagine this is difficult but if they are visiting feeders regularly, would some index of association in the pre-breeding season be available beyond just when they first meet? It seems like this would really strengthen the study.

RESPONSE: We thank the reviewer for the comments. In response to the reviewers opinions on whether this warrants publication in Proceedings B, we -first of all- hope that the editor agrees it is potentially suitable, and we are very confident that this study will have a broad appeal to the readers, as well as encourage others to further explore this important topic. In particular, the reviewer themselves states that this study is indeed novel, as it is the first study to show direct links between meeting time and breeding success in wild, and also as it is one of the only three studies so far conducted on the pair familiarity prior to first breeding and breeding success.

Further, based on the comments from the associate editor and both reviews, we have thoroughly revised the manuscript to make clear that in our study system we definitely know (previous work on the same data) that members of a future breeding pair associate together more consistently and generally much more frequently than other flockmates. This is further reflected in our additional supplementary analysis we have now added to the manuscript. Specifically, we use the SRI scores and several of its derivatives, to show that results are robust to the measure of the association with the partner, which shows that it is meeting time and prolonged association that leads to increase in breeding success. Please also see responses to the associate editor.

COMMENT: The title is informative but quite a mouthful, although I do like the first part. Is it important to mention their mating system? Why not just say “Familiarity breeds success: early pre-breeding meeting has breeding benefits in great tits”?

RESPONSE: We have changed the title to: Familiarity breeds success: pairs that meet earlier experience increased breeding performance in a wild bird population

COMMENT: Pg 1 of intro focuses on the mate familiarity hypothesis, but there is also the reproductive performance hypothesis (alluded to later), which suggests that more experienced parents have a fitness advantage. It's true that this applies less to monogamous species, but monogamous species also divorce. I also found that the intro was almost exclusively focused on birds for quite obvious reasons. Nevertheless, there are papers on other system alluding to the role of familiarity and it's influence on mating. For example, see Leu et al. 2015 Ethology for lizards. I do appreciate that space is a constraint and just suggest a few more sentences to expand beyond birds.

RESPONSE: We now acknowledge that other factors can also influence breeding success, and this now reads: 'The increase in breeding success partly arises because of the 'mate familiarity effect' [1,4,5], even when accounted for confounding factors, such as age/experience related increases in reproductive parameters [1, 6], as familiar partners improve coordination, cooperation and responsiveness [7,8].'

We have now also added examples and references to other study systems that address the issue of familiarity. We have further used these to discuss what familiarity can represent (please see the response to the associate editor, point 2)

COMMENT: 17 delete "to"

RESPONSE: Done

COMMENT: 21-25 be explicit about the relationship between meeting date and whether a pair stays together or divorces.

RESPONSE: We have now made it clear that there is no influence of meeting time on divorce

COMMENT: 68 lower probability

RESPONSE: Corrected

COMMENT: 75 I know that great tits are very well studied, but I suggest greatly expanding this paragraph (at least by 2-4 sentences). Given the nature of this study, as a reader, I would like to know some basics about the breeding biology of great tits in the Wytham population. This includes the age they first breed, when do they typically pair up to breed, when do they lay eggs (date range) and is this variable between years in the same pair. Can it depend on a range of environmental variables that can be briefly mentioned? And say something about their mating system. As I understand it from other work, they are socially monogamous but roughly half of broods contain EP offspring? These mating system and life history variables will help the reader place this study in context.

RESPONSE: We have added more details on the study species for readers unfamiliar with the system. Further, some of the relevant details are provided later in the Methods (e.g. that breeding experience and age influence breeding success).

COMMENT: 76 short lived (give a range or some other measure, in years)

RESPONSE: We have added survival rates of adults.

COMMENT: 109-111 I don't understand this. Why are they different (first 3 vs last 3 winters)?

RESPONSE: This is dependent on the differences in the data collection set up in these two set of winter. We now add more detailed description of the protocols to the *Data collection set-up* section, and add an additional sentence to the part the reviewer is referring to make it clear why these time periods were selected.

COMMENT: 112 Is "preseason bonding" the same as preseason meeting? As I understand it you established when unfamiliar pairs first met prior to the breeding season. This might require a reword.

RESPONSE: We have rewritten this part, so there is no mention of any of the two terms. Based on the comments from associate editor and the other reviewer, we now avoid using the term bonding, as we cannot make an objective decision when a bond is strong enough to be called a pair bond (although this bond is stronger than the bond with other individuals). We rather opted to use a more neutral language to only state that birds do associate from the meeting time on, and that they do so more frequently than any other birds.

COMMENT: 114 Further,

RESPONSE: We have rewritten the whole section, thus the comment is now irrelevant

COMMENT: 205-206 I would include a sentence explaining why these species were selected. Saying they are closely related is not enough of a justification. I would assume they are similar in their social and reproductive behaviour?

RESPONSE: We have added details on this analysis to the supplement. Because the species are closely related (and by it have relatively similar reproductive biology and breeding behaviours), it is likely that the evolution has selected for the same components of fitness to be the main triggers of divorce.

COMMENT: 230-231 I didn't get this from the previous sentence?

RESPONSE: We have revised this section to be clearer.

COMMENT: 233 what about the type of pair?

RESPONSE: We have revised this section to be clearer about the results. Further, this has been showed in the Fig 3.

COMMENT: 238)

RESPONSE: Corrected

COMMENT: 255-256, 257-258 yes, but in what direction? Can you reword to explicitly give the relationship?

RESPONSE: These estimates are all non significant, likely due to a small sample size. We add this to the Results.

COMMENT: 266-268 this seems to be in contrast to results?

RESPONSE: Although model selection did not provide a clear support for only one model, the term meeting time has not reached statistical significance in any of the models (maybe because of the small sample size). Thus, we do not have any evidence for the effect of meeting time on fidelity. We are clear to report this consistently in the text when discussing this relationship.

Again, we thank reviewer 2 for their positive comments and their helpful points made here.

Appendix B

Response to Referees and manuscript with track changes

RESPONSE TO REFEREES

Associate Editor, Comments to Author:

This re-submitted manuscript was reviewed by one of the previous reviewers (#1) and by one new reviewer (new #2). Both I and reviewer #1 found the manuscript to be substantially improved over the first submission, and were convinced by the authors' argument that their methods adequately measure time since pair-bonding and not simply time since first detection in the same flock (which was a major criticism of the first submission). Reviewer 2 also has an overall positive impression of the paper. However, both reviewers also make a number of useful suggestions for further improvements. Based on these comments and my own reading of the paper, I think that the most significant points can be summarized as follows:

RESPONSE: We are glad to hear that our manuscript is now clearer. We thank the Associate Editor and the reviewers for pointing these considerations out in the previous round of review. We have addressed the comments from the Associate Editor and both reviewers, as detailed in the point-by-point responses below. We have also added the statement on the use of animals to our methods section, and provided links to the datasets and code in the data availability statement.

COMMENT: First, the overall effect of pairing date on lay date seems robust. It emerges in both data sets (2008-2010 and 2012-2014) and happens regardless of pair type (experience of the male or female in the pair). However, effects on other components of reproduction, such as hatching and fledging success, get harder to interpret as the data are separated by category. My and reviewer #1's comments on Figure 3 reflect this difficulty: We both question whether the sample sizes in different categories are large enough to reflect meaningful differences. Instances when results differ between the two data sets are perhaps the most challenging to understand, because it is not clear (to me, at least) whether results differ due to the different methods employed in the two periods, biological differences between years (i.e. due to weather), or just random chance. (For example, the statement in the abstract that earlier-meeting pairs had higher fledging success "in 3 of 6 years" is not easy to interpret.)

RESPONSE: We have revised our statistical method approach based on the comments from Reviewer #1 and the Associate Editor (i.e. in line with Arnold, T. W., 2010), as well as consulting the literature on over-parameterisation of models (Harrell, F.E: Regression modelling strategies. 2010). Based on the latter, we considered only those model structures where the number of model parameters was lower than the sample size/10 (as the non-conservative upper boundary). This equals to 14 parameters for the model selection on the first dataset, and indicates that some of our previous models were over-parametrized, and thus, as the Associate Editor points out, did not reflect meaningful differences between datasets. We have added explanation of these rules for model selection to our Method section, and we cite the appropriate literature for this. After we revised our approach, the effect of meeting time on laydate remained robust, which is encouraging as this has always remained our primary finding. However, the effect of meeting time on hatching and fledgling success in the first dataset was not supported, which is different from our first analysis but in line with our initial conclusions that these particular relationships were not consistently found across the data. We have also adjusted our discussion in line with these more consistent, robust results, and all the changes are marked.

COMMENT: Second, the results of top models need to be reported in the supplementary tables, as well as in the main text for the most important/significant findings. As reviewer #1 notes, this is necessary to evaluate the significance of predictors that are included in top models. Reviewer #1 also argues for removal of non-significant parameters from top models. While I do not think that this is necessary, I do agree that it is necessary to present the estimates, confidence intervals, and p values for each predictor retained in the top models in order for the reader to evaluate which ones are meaningful. Reviewer #1 also notes that when two competing models have similar AIC scores, the model with fewer terms is typically thought to be more informative. The rule of thumb that I usually follow is that if two models are within 2 AIC units, the model with fewer parameters is more informative (see Arnold, T. W. (2010). Uninformative parameters and model selection using Akaike's Information Criterion. *The Journal of Wildlife Management*, 74(6), 1175-1178.)

RESPONSE: We now report the full results of the model selection (including step 2 and 3) in the Supplement. We also add Table 3 with the results of the best laydate models (both datasets) to the main text, and the tables with the results of best supported models on other

components of breeding success to the supplement (S18-S21). As explained in our earlier response (to the comment above), we now also follow the suggested rule-of-thumb (we add these changes to the Methods section along with the supporting references).

COMMENT: Third, the presentation of results in the main text and especially in the supplement needs a lot of work to improve clarity. The captions of most of the supplementary figures are impossible to understand without reference back to the supplementary text and main text, so they add little to the readers' understanding (as currently presented). Figure S8 appears to be identical to Fig 1 in the main text. There are also many typos in the results and captions that add to the overall impression of carelessness. I would recommend going through the results and supplement with a fine-toothed comb to correct these errors and adequately explain the supplementary figures.

RESPONSE: We have now gone through these sections to improve the clarity. Figure S8 (now Fig S5) and Fig 1 are not identical as the Fig S8 indicates the effects we have shown in this study (red arrows) - we now ensure this is clear in the figure legends.

COMMENT: Fourth, Reviewer 2 suggests several references to strengthen the introduction, which should be incorporated.

RESPONSE: We have added these references (all but one, please see the response to the Reviewer #2) to the introduction.

I have made a few line-by-line comments as well, which are below.

COMMENT: 20: “experienced higher fledging success in 3 of 6 years”

RESPONSE: This has been removed as the result is no longer supported (see our previous responses). We now have added the sentence: ‘However, clutch size, number of hatched and fledged young, and hatching and fledging success were not influenced by the meeting time.’

COMMENTS: 103: please clarify that this is annual survival (I assume?). How long is average life span in this population? Is the average of 7 chicks fledged for successful nests only, or all nests? Finally, it is probably best to specify that this population is non-migratory and does not undertake local movements during the winter.

107-108: it is unclear whether “most birds recruit into the breeding population in their first year” refers to all one-year-olds. What is the survival rate from hatching to one year, and what proportion of those surviving at 1 year breed in their first year? 110-112: what is the number of nest boxes and breeding pairs in the population? Are the nest boxes protected against predators? I realize that this is an extremely well studied population, but please give basic info for readers so they do not have to refer to other papers.

RESPONSE: In response to these comments, we have added more information about the study system at the beginning of our methods section (lines 101-118)

COMMENT: 122-123: what were the sample sizes of marked individuals?

RESPONSE: We have now added this information to this part of the methods (and state “...with 4023 unique great tits detected over the 6 study years considered here...”)

COMMENT: 201-203: If I understand this correctly, a simpler way of explaining might be that the first step of model selection was full model subsetting in which candidate models were generated from all combinations of predictors and their two-way interactions.

RESPONSE: We have changed this part of the text to be clearer, as suggested. However, we note that only those combinations leading to $\max k = \text{sample size}/10$ parameters were considered.

COMMENT: 203-205: Please clarify that this means that meeting time was categorized monthly for the first data set, and weekly for the second data set. This is not really obvious to the reader until later in the results, when it comes as a surprise.

RESPONSE: Clarified. We avoided use of the word ‘categorized’ as it might imply it is categorical, rather than continuous, variable.

COMMENT: 259: This too needs similar clarification -- just a reminder that meeting time was categorized either by month or by week, depending on the data set.

RESPONSE: Clarified

COMMENT: 277: Fig 2 – why is the data separated by year in 2a, but not in 2b?

RESPONSE: Because model selection did not support the influence of year in the last three winters.

COMMENT: 286-289: I think this must be referring to Fig 3b, not 2b. Fledging success is not shown in Fig. 2b. In fact, Fig 2b refers to the later data set, and this paragraph otherwise seems to be discussing the earlier dataset.

RESPONSE: This section has been removed as the results were not supported after we have changed our approach to model selection, and the maximum number of parameters.

COMMENT: Correct the typos in both figure captions.

RESPONSE: Corrected

COMMENT: Overall, results section needs to have the actual results of the top models presented (estimates of each parameter, confidence intervals, and p values)

RESPONSE: We have added Table 3 to present the results of the top laydate models in the main text, and Tables S18-S21 to present the results of the top models for other breeding success components.

COMMENT: 304: Fig 3 – some of these patterns are quite messy and hard to interpret, particularly 3a. Do these lines reflect anything real, or just odd patterns resulting from small sample sizes? Unless these results have a more compelling explanation that I perceive, I'd be in favor of simplifying the results and eliminating this figure. In 3b, the overall negative effect is obvious (lower fledging success with later pairing) but the differences between categories are more confusing. Why do the most experienced pairs (yellow line) have lower fledging success than the rest? Again, it would be worth considering whether the sample sizes allow for meaningful comparisons between categories. If the power is not sufficiently high, it might be better to pool the data.

RESPONSE: This figure has now been removed, as the results are no longer supported. Please see the responses to the first two comments.

Thank you very much again for your positive comments and detailed suggestions.

Reviewer(s)' Comments to Author:

Referee: 1

Comments to the Author(s).

COMMENT: This is my second reading of this ms. I find the clarification about how to differentiate between a pair and just another flock member (and the additional analyses to support this) helpful, thank you. I do however have other areas which I think are still not clear. Also, I found the response to the questions about model selection unclear. On consulting your AIC tables in the supplementary material, I think you have done these as expected for model selection, and so your response, while unclear, seems to not have affected the AIC output. However, it seems some AIC tables are missing? And why are effect sizes and confidence intervals not given for terms in top models? These are important to determine the effect of different predictor terms on data patterns.

REPOINSE: We thank reviewer for repeatedly reading and commenting on our manuscript. We address the reviewers remaining concerns, as outlined point-by-point below. We now provide the full results of the top models in the main text (Table 3) and supplement (Tables S18-S21).

COMMENT: L 24: 'opens a new area in the study' – what is the new area? This statement is a bit vague/ambiguous and I suggest deleting it.

RESPONSE: We have removed this.

COMMENT: L 51: typo 'each-other'

RESPONSE: Corrected

COMMENT: L 55: typo, should read 'reinforcing'

RESPONSE: Corrected

COMMENT: L 69: add reference

RESPONSE: Added

COMMENT: L 86: replace 'probably' with 'probability'

RESPONSE: Corrected

COMMENT: L 103: are these annual survival rates?

RESPONSE: Yes, added for clarification.

COMMENT: L 105: are these numbers per nesting attempt? Or per season (multiple nesting attempts?)

RESPONSE: Added 'per nest'.

COMMENT: L 121: typo: should read 'transponders'

RESPONSE: Corrected.

COMMENT: Methods: so data collection is reliant on RFID tagged birds visiting feeders: but how often do birds visit feeders? Do some of them never visit feeders?

RESPONSE: We previously provided the proportion of birds which were tagged and visited the feeders over the winter season but we did not make this clear. We have now reworded this information so that it reads "...In this study system, an estimated 82% of the primary population is marked with PIT-tags and visit the recording devices over the winter season..."

COMMENT: L 159: this information is important: the sum of the times when they were observed at all, since it affects how a pair is defined. Feeders give a snapshot in time, and so is a pair defined as being at the same feeder in a weekend for 10 seconds or less? As in, please give the context for 'time observed at all'. How often were they typically observed for per week/month on average? L165 suggests observations include individuals that were detected at a feeder just once, suggesting observations are only a few second long?

RESPONSE: The reviewer is correct in stating that the feeders give a snapshot in time of the social structure, and this is exactly what our methodology hopes to achieve. These birds roam in flocks freely through the woodland over the winter period, so we aim to gain snapshots of the social structure every weekend. We did not aim to actually quantify the amount of time these birds are specifically at a particular feeder together. Indeed, these are sunflower seed feeders and the birds rapidly take turns at collecting seeds and then processing the seeds in the surrounding vegetation. Rather, we aimed to gather knowledge of their general flocking associations through this large-scale snapshot sampling approach. We have now clearly stated this in the text, and we refer to prior work that demonstrates how this approach is useful for gathering information on general social structure and pair bonding (we state "...Our previous work has demonstrated that our data collection methods (sampling snapshots of social structure throughout the winter) in combination with creating SRI social networks from the underlying gambit-of-the-group approach are good representations of social associations and bonds between individuals (Psorakis et al. 2015; Firth et al. 2018)...")

COMMENT: L 174: how does this rate (29 and 9) compare to the population average of associations between non-breeding birds for context? As in, is this rate unusual?

RESPONSE: We believe the reviewer is asking whether it is unusual for two birds who are not paired together to not meet during the winter. The answer to this is that most birds never meet, as we detect ~750-~1000 great tits per year across the woodland, and most birds have an average of ~60 unique flockmates in each year.

COMMENT: L 187: this statement is vague, please clarify 'we compared the estimates of the effects'

RESPONSE: We have changed this section (see response to comments from the Associate Editor). The sentence referred to here has now being removed.

COMMENT: In response to my question about the model selection, the authors have answered as follows: ' . The issue here is that it can happen that removing the statistically non significant predictor can also change the estimate of the significant predictor. In our analysis we have opted for a second choice, where we compare plausible model structures, and select the best supported model(s) based on their AIC. These models will indeed contain some insignificant terms, but removing these terms can further influence the estimate of the remaining effects.'

I disagree with this: the removal of a non-significant predictor should not change the estimate of the significant predictor. This might happen if the two terms are correlated, and hence why correlated terms should not be included in the same model together. Model selection involves comparing between different hypotheses. Where two models have similar AICc, the simpler model, with less terms, is typically chosen, since the more complicated model likely has non-significant terms contained within it. Important terms can be determined by determining whether the standardized confidence intervals intersect zero A very helpful guideline for model selection can be found in this widely cited recent paper Harrison et al 2018: A brief introduction to mixed effects modeling and multi-model inference in ecology.

RESPONSE: Based on the comments from the Reviewer and the Associate Editor, we have decided to consider the model with the lowest AIC and with the lowest number parameters

amongst the models with similar AIC (<2 AIC) as the best supported (even if this model includes some non-significant parameters). The results (estimates) of these models are presented in tables S18-S21.

COMMENT: Methods: GLMMS are described but the use of random terms to account for repeated measures not stated, even though they should be present.

RESPONSE: Only very few birds in both datasets were recorded breeding in more than one year (9% of individuals). Thus, we did not include the random effects term of this individual-level component. We have added this information to the methods.

COMMENTS: L 268: please put more biology into this. It currently reads as ‘supported the effect of meeting month on laydate. This is confusing. Supported what effect? An hypothesized positive or negative effect? Reading this sentence, it is unclear whether meeting month advanced or delayed laydate. This problem is in L 270 as well: significant predictors of hatching and fledgling. OK, they are significant: in what direction?

COMMENT: L 274-275: refer to the table that supports this statement.

COMMENT: L 282-289: this issue of not putting the biology into the results statements affects this paragraph too: terms are stated as important, but directionality of effect is not given.

RESPONSE: In response to these comments above, we have considerably changed the results section, and agree this increases biological focus.

COMMENT: Fig 3: the interaction in (a) is very difficult to understand. What is the sample size for the four different pair types. The sample size to detect this type of interaction with sufficient statistical power is important.

RESPONSE: Fig 3 has been removed (based on the input from the Reviewer and Associate Editor, we have revised our analysis - please see the detailed answer to the first two AE comments).

COMMENT: Top model sets for main results should be present in the main text, not supplemental material. Full model sets are ok for supplemental material, but I feel strongly that the reader should be able to see the top model sets in the main text. It allows the reader to see model support, effect size and directionality, and variance.

RESPONSE: We provide the detailed model output for the main models (laydate) in the main text in Table 3, and for all other breeding components in the Supplement (Tables S18-S21).

COMMENT: Supplemental material: I can only find two AIC model output tables, yet the authors conducted more analyses than this? Where are the other AIC output tables?

RESPONSE: We have added Step 2 and step 3 to these tables – these steps are parts of the main analysis. However, we did not add the AIC output tables for the sensitivity analysis. This would considerably lengthen our (already long, 24 tables + text) supplement, especially because this analysis did not lead to any change in the overall results. Further, all of our data and the code (including the code for the sensitivity analysis) are provided in Dryad, so all the results can easily be repeated.

Referee: 3

Comments to the Author(s).

The authors have done a nice work having into account the different comments by the referees. A critical point was to clearly show that members of a future breeding pair associate together more consistently and generally much more frequently than other flockmates, and I think that authors have provided a solid argument to support their statement. I also appreciate the effort of authors to show that it is meeting time and prolonged association that leads to increase in breeding success.

I also appreciate the effort by the authors “to put (their) study into a wider context of familiarity that (they) had previously focused on”. I really think that the topic of “familiarity” has been for years unappreciated. However, within this wider context I miss some reference to previous significant work on the topic. For instance, Cheetham et al. (2008) nicely reviewed the topic on the importance and advantages of familiarity with potential mates to increase

fitness. Two more recent papers in Proceedings have shown that individuals show a marked preference to pair with familiar individuals (i.e. from their social group), which may elicit a so strong attraction as traditional ornaments generally assumed to be used in mate choice (Senar et al. 2013; Thunken et al. 2012). They also discuss on the topic. I think these papers should be used to improve the discussion or set the INT on the important and under-appreciated role of familiarity in mate choice. I think this would give to the paper a wider interest.

In the 70's, Saitou wrote a series of very nice papers on flock dynamics along the year in Great tits, and one of the papers specifically was devoted to pairing dynamics within the flocks (Saito 1979). I think you should have a look to this paper and cite it. Saito found that pairs in Great tits were formed from February on, rather than in autumn, as you found. Previous work on great tits, in addition to Saito (1979), also reported that in spite that individuals could flock together in winter, pairing was really taking place in spring (Hinde 1952; Kluijver 1951). I think this merits some discussion. In medaka fish it has been shown that a close association between the members of the "future" pair allows to increase familiarity between them, which is later basic to increase pairing success (Yokoi et al. 2016), but "pair bond" as such, do not takes place at this period of social association but later. I wonder whether in Great tits it could happen the same. Hence, I also would appreciate some discussion on the topic.

I hope these comments help authors to give their very nice paper a wider interest.

Juan Carlos Senar

RESPONSE: We thank the reviewer for their encouraging words. We have read the suggested references with interest, and have revised our Introduction to incorporate these references, and to also introduce the topic of familiarity better. Our changes can be traced via track changes. We only did not include the reference Thunken et al. 2012, as this seems to be more about genetic relatedness of individuals.

Appendix C

RESPONSE to REFEREES and TRACK-CHANGES MANUSCRIPT

Response to Referees

Dear Dr Culina:

Your manuscript has now been peer reviewed and the reviews have been assessed by an Associate Editor. The reviewers' comments (not including confidential comments to the Editor) and the comments from the Associate Editor are included at the end of this email for your reference. As you will see, the reviewers and the Editors have raised some concerns with your manuscript and we would like to invite you to revise your manuscript to address them.

RESPONSE: Thank you for the invitation to revise the manuscript. We have addressed the Editors comments as we outline below (we could not find any issues raised by the reviewer, but we thank them for their initial comments and for their positive assessment here).

Associate Editor
Board Member: 1

Comments to Author:

This well-written and well-presented paper on the fitness consequences of meeting time in pairs of great tits is a revision of an earlier submission. The authors have re-done the statistical analysis as recommended, which has changed the results somewhat. The authors now find that pairs that meet earlier in the winter experience increased reproductive success in the breeding season, but this is due to the effect of lay date. Pairs that meet earlier tend to lay earlier and to have larger clutches, which leads to a higher number of offspring fledged.

RESPONSE: We thank the Associate Editor for these positive comments.

Although the revision is much improved (and the authors have supplied much more information on the statistics in supplementary tables, which is great), I still have two remaining concerns. First, as currently written, the abstract implies that pairing time affects lay date but does not affect subsequent reproductive success. This makes the title and conclusions seem unsupported. After reading the ms, I realized that pairing time actually does affect reproductive output, but it does it indirectly through the effect on lay date (in other words, when controlling for lay date, meeting time does not affect rs). This is important because it means that there is no additional advantage to a longer pre-breeding period with a social mate (for example, increased pair coordination). The introduction similarly fails to make it clear that it is already known that earlier lay date leads to increased reproductive output (this needs to be explained and earlier studies cited), and the results mention it only superficially. Given that this is a crucial link in the argument that meeting time has subsequent fitness effects, the authors need to explain this more clearly and not assume that the readers are already familiar with the previous work. The below line-by-line comments highlight areas where this information is missing in the ms.

RESPONSE: Thank you for pointing this out. We have now make this point clear: we find that meeting time influences the laydate directly, but other components indirectly (via the effect of the laydate, as earlier laying pairs have larger clutches and higher hatching and fledgling success). This second finding (that laydate influences breeding success) has also been detected in the previous studies. We now make this more clear throughout as recommended by the AE. Please see details of our revisions in our responses to the line-by-line comments below.

Second, the central finding of the paper is that earlier meeting time leads to earlier lay date, illustrated in Fig. 2. However, it is not clear to me why meeting date is not standardized within years the way that lay date is. It is difficult to interpret a figure that shows a calendar date on the x axis and a standardized date on the y axis. Of course, I understand why lay date is standardized -- because in some years laying is earlier than in other years, and you are interested in the timing of laying of an individual pair relative to the rest of the population in that year. But it seems like the same logic should apply for meeting time: you are not necessarily interested in the calendar date, but in whether the pair met earlier or later in the year than other pairs in the population.

RESPONSE: Thank you for pointing this out, we had previously considered this point ourselves too and opted for using the actual timing within each year, to capture the actual time each winter and to allow for relevant and intuitive interpretation of the results in relation to the actual months. However, we also do agree that scaled timing is indeed potentially important for the reasons pointed out here. Fortunately, the correlation between real meeting time and standardised meeting time is >0.9 in all years, so these are almost equivalent. Nevertheless, we have also now run an additional analysis on the laydate where the meeting time is standardized (added lines 250-252), and add the additional details on the model selection to the Supplementary tables S3 and S11 – model selection gave support to the same model structure as the model selection where the meeting time is not standardized, apart from the model selection on the second set of years, where the effect of meeting time was linear (rather than cubic, as in the model selection using non-standardized meeting time) . We provide the estimates of these models in Supplementary table S2). We further provide supplementary Figure S5 (to which we now refer to in the Results section) with plots that have standardized meeting time. We have kept real calendar meeting time in the main MS having meeting time as the real value. Again, we feel it is easier to interpret (easier to connect to the real months, and thus the natural history of these birds), including our description of the results in the main text. Further, we believe that this key figure (using real time) is easy to interpret, and we hope it is so we generally an intuitive representation of the results. However, if Editors find that the figures with the standardized meeting time should go into the main MS, and the figures with the non-standardized meeting time in the Supplement, we can do so, but would prefer to stick to the real-time measures especially given as it is almost equivalent anyway. Slopes and the CI are almost identical in both supplement and main MS figures.

I thank the authors for their changes to the statistical analysis and would welcome a revision if the two above points can be addressed.

RESPONSE: Thank you again for your comments, and we believe we have address the two main points.

Line-by-line comments:

19-21: This needs to be clarified. If I understand the results correctly, meeting time influences reproductive output indirectly, because pairs that meet earlier also breed earlier, and pairs that breed earlier have larger clutches

RESPONSE: We now clarify this: 'Clutch size, number of hatched and fledged young, and hatching and fledging success were not influenced directly by parents' meeting time, but indirectly: earlier laying pairs had larger clutches (that also produce higher number of young), and higher hatching and fledging success.

76-78: This needs to be expanded to explain that it is already known that earlier laying leads to fitness benefits, and to explain what those benefits are and cite the studies that have found them. Essentially you are testing three hypotheses: 1) that earlier pairing does not change subsequent reproductive timing or outcomes (null); 2) that earlier pairing leads to reproductive benefits directly, via increased coordination or improved parental care; or 3) that earlier pairing leads to reproductive benefits indirectly, via earlier laying.

RESPONSE: We have now increased the amount of information on this point. Specifically, as well as stating how laydate is well known to be under strong selection due to strongly affecting later components of breeding success (lines 69/70 - and providing four references to support this claim), we now also add : 'Previous studies have already showed that earlier laydate generally translates into higher breeding success and reproductive output [21-24] ' to the lines 80-81.

279-281: But did lay date itself affect reproductive success (clutch size and hatching and fledging success)? It's still not clear to me whether earlier laying translated into reproductive benefits, or whether other studies have found this pattern but it was not actually recapitulated here.

RESPONSE: We agree it is important to be very clear about this point and we now clarify our findings further here by adding 'had a direct effect on the standardized laydate'. We also move the paragraph (previous lines 303-307) to the beginning of the Results section to clarify earlier on that we have also shown that laydate affects later components of fitness. Our results showed (and this is also in line with results of previous studies) that earlier laydate leads to a larger clutch size and higher hatching and fledging success. Our study has also showed that larger clutches produce higher number of hatched young, and that the larger number of hatchlings translates into higher number of fledglings (the details of the model estimates and sizes of these effects are provided in Tables S18-S21).

300-304: This section seems to say that lay date did have knock-on effects of reproductive output, but there are no actual results here. Table S18 shows that the best-supported model for clutch size included lay date as the sole predictor.

RESPONSE: We have now moved this section at the beginning of the results, and explain the results more clearly (in line with the response to the above point too).

336: This needs to be explained in the introduction.

RESPONSE: We have now done this as outlined in our responses to the above comments.

376: the phrasing here of “direct positive benefits” is a bit confusing, given that you have described the effect of meeting time on reproductive success (mediated by lay date) as an indirect effect in Fig. 1 and the introduction. It would be clearer to say that meeting time influences lay date, which in turn has positive effects on reproduction.

RESPONSE: Text changed to read: ‘These benefits appear to arise via positive effects of earlier meeting time on the laydate (earlier laydate), which in turn increases breeding success’

Thank you again for your comments and helpful suggestions.

Reviewer(s)' Comments to Author:

Referee: 3

Comments to the Author(s)

The authors have done a good job reviewing the paper.

RESPONSE: Thank you for your initial comments, and we are pleased to hear that the revision was appreciated.